# *Trans*-eQTL mapping in gene sets identifies network effects of genetic variants

## Graphical abstract

## Authors

Lili Wang, Nikita Babushkin, Zhonghua Liu, Xuanyao Liu

## Correspondence

xuanyao@uchicago.edu

## In brief

Establishing a comprehensive map of *trans*-gene regulatory effects is a critical step toward understanding complex trait and disease genetics. However, detecting these effects is extremely difficult due to statistical and computational challenges. Wang et al. developed a powerful tool, *trans*-PCO, to detect high-quality *trans*-genetic effects on gene networks, which opens up new opportunities to learn the impact of trait-associated loci on gene regulatory networks.

## Highlights

- *Trans*-PCO outperforms existing methods by finding more high-quality *trans*-eQTLs

- *Trans*-PCO offers a map of *trans* regulation of gene networks and biological processes

- Functional annotation of gene modules helps functional interpretation of *trans* signals

- *Trans* effects via regulatory networks and pathways reveal the mechanism of trait loci

Wang et al., 2024, Cell Genomics 4, 100538
April 10, 2024 © 2024 The Author(s). Published by Elsevier Inc.

CellPress

# *Trans*-eQTL mapping in gene sets identifies network effects of genetic variants

Lili Wang,[1,2] Nikita Babushkin,[2] Zhonghua Liu,[3] and Xuanyao Liu[1,2,4,5,*]
[1]The Committee on Genetics, Genomics and Systems Biology, University of Chicago, Chicago, IL 60637, USA
[2]Department of Medicine, Section of Genetic Medicine, University of Chicago, Chicago, IL 60637, USA
[3]Department of Biostatistics, Columbia University, New York, NY 10032, USA
[4]Department of Human Genetics, University of Chicago, Chicago, IL 60637, USA
[5]Lead contact
*Correspondence: xuanyao@uchicago.edu

## SUMMARY

Nearly all trait-associated variants identified in genome-wide association studies (GWASs) are noncoding. The *cis* regulatory effects of these variants have been extensively characterized, but how they affect gene regulation in *trans* has been the subject of fewer studies because of the difficulty in detecting *trans*-expression quantitative loci (eQTLs). We developed *trans*-PCO for detecting *trans* effects of genetic variants on gene networks. Our simulations demonstrate that *trans*-PCO substantially outperforms existing *trans*-eQTL mapping methods. We applied *trans*-PCO to two gene expression datasets from whole blood, DGN (N = 913) and eQTLGen (N = 31,684), and identified 14,985 high-quality *trans*-eSNP-module pairs associated with 197 co-expression gene modules and biological processes. We performed colocalization analyses between GWAS loci of 46 complex traits and the *trans*-eQTLs. We demonstrated that the identified *trans* effects can help us understand how trait-associated variants affect gene regulatory networks and biological pathways.

## INTRODUCTION

More than 90% of genome-wide association studies (GWASs) loci are located in noncoding regions of the genome and are thought to affect human traits by regulating gene expression.[1–5] Nearly all of the studies to date have focused on understanding the effects of trait-associated variants on gene expression in *cis*, which only include effects on genes that are near the associated loci. However, multiple lines of evidence suggest *cis*-regulatory effects capture only a small proportion of the heritability of complex traits and diseases. We previously hypothesized that *trans*-expression quantitative loci (eQTLs), despite having very small effects on each individual gene, may cumulatively account for a large proportion of trait variance.[6] Indeed, our modeling indicates that *trans*-eQTL effects account for twice as much genetic variance in complex traits as *cis*-eQTL effects.[6] Thus, establishing a representative map of genetic variants and their *trans* effects is a critical step toward understanding complex trait and disease genetics.

Two major challenges have precluded *trans*-eQTL discovery. First, *trans*-eQTL mapping is extremely prone to false positives due to mapping errors that cause short sequences to map to homologous regions of the genome.[7] The second challenge is by far more difficult to overcome: *trans*-eQTLs are challenging to detect compared to *cis*-eQTLs because (1) they have much smaller effect sizes than *cis*-eQTLs[6] and (2) a genome-wide search of *trans*-eQTLs involves a huge number of statistical tests, resulting in a heavy burden of multiple testing corrections.

Previous work suggests that *trans*-eQTLs generally affect the expression levels of multiple genes.[8,9] The co-regulation and co-expression patterns of genes driven by *trans*-eQTL have long been recognized. There are a few studies that aimed to identify *trans*-eQTLs of co-expressed genes. For example, Rotival et al.[10] used independent component analyses to identify co-expression gene sets, and subsequently tested for the enrichment of *trans* signals in the gene sets by hypergeometric tests. More recently, Kolberg et al.[11] tested associations between SNPs and an "eigengene" (essentially the primary principal component, PC1) of gene modules that captures the co-expression pattern. Nonetheless, PC1 has very limited power at identifying genetic effects (see below). Dutta et al.[12] leveraged canonical correlation analysis to identify *trans* associations between multiple disease-associated SNPs and multiple genes by integrating with GWAS signals. However, the method has different goals from identifying *trans*-eQTLs of multiple genes in specific tissues (e.g., it is useful for identifying "core"-like disease genes and processes for a specific disease; see below).

Our main goal was to develop a method for detecting *trans*-eQTLs associated with multiple genes in a gene module by using multivariate association. Multivariate association methods tend to be more powerful than univariate association methods. Detecting *trans*-eQTLs of gene modules containing multiple co-regulated genes can also potentially improve power by reducing multiple testing burdens, because the number of tested gene modules is much less than the number of genes. However, there are caveats. First, sequence similarity among distinct genomic

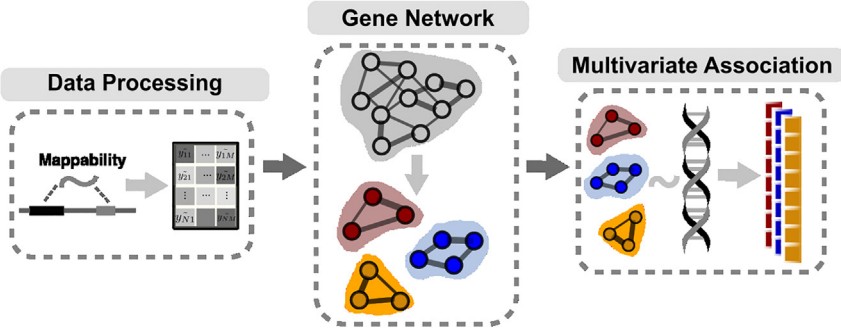

**Figure 1. Three main steps of *trans*-PCO pipeline**

First, *trans*-PCO preprocesses RNA-seq data to reduce false positive *trans*-eQTL associations. Second, genes are grouped into gene sets, such as co-expression modules or biological pathways. Lastly, *trans*-PCO tests the association between SNPs and gene sets by using PCO.

regions can lead to severe false positive discovery issues in *trans*-eQTL mapping.[7] This is especially problematic in mapping *trans*-eQTLs of co-expression gene modules because genes can be falsely clustered due to sequence similarities.[7,13] Second, the naive way of using a single component, such as the first gene expression PC, to represent the gene modules can significantly reduce power. Although the PC1 captures the largest amount of total variance in gene expressions, it can be powerless in detecting significant associations than higher-order PCs.[14,15]

To combat this, we propose *trans*-PCO, a flexible approach that uses the PC-based omnibus test[15] (PCO) to combine multiple PCs and improve power to detect *trans*-eQTLs. *Trans*-PCO also carefully filters sequencing reads and genes based on mappability across different regions of the genome to avoid false positives due to multimapping.[7,16,17] By default, *trans*-PCO uses gene sets identified by weighted gene co-expression network analysis (WGCNA),[18] which clusters co-expressed genes by using the correlations of gene expression levels. It also accepts user-defined sets—for example, genes that belong to the same Gene Ontology,[19] Kyoto Encyclopedia of Genes and Genomes pathway,[20] or protein complex.[21]

We applied *trans*-PCO to gene expression data from the Depression Genes and Networks study[16] (DGN, sample size N = 913) and the eQTLGen study[9] (sample size N = 31,684) to identify *trans*-eQTLs associated with co-expression gene modules and well-defined biological processes in whole blood. All *trans*-eQTLs that are associated with gene co-expression networks and biological pathways can be found at http://www. networks-liulab.org/transPCO.

## RESULTS

### Overview of the method

The *trans*-PCO method consists of three main steps (Figure 1). First, *trans*-PCO preprocesses RNA sequencing (RNA-seq) data to reduce false positive *trans*-eQTL associations due to read multimapping errors. Specifically, *trans*-PCO removes all of the sequencing reads mapped to low mappability regions of the genome (mappability score < 1; STAR Methods) before profiling gene expression levels. These procedures substantially reduce the occurrence of false positive *trans*-eQTLs due to sequencing alignment errors.[7,17] When only summary-level data are available (e.g., eQTLGen dataset[9]), *trans*-PCO dynamically excludes from the module any genes that are cross-mappable to genes within 100 kb of the tested SNP.

Second, *trans*-PCO groups genes into clusters. By default, *trans*-PCO determines the gene groupings by using WGCNA[18] to identify co-expression modules from gene expression levels (STAR Methods). We remove covariates and confounders (STAR Methods) from gene expression levels before grouping gene modules. This step is necessary to ensure that the gene modules are not primarily driven by confounding factors. *Trans*-PCO also allows customization of the gene groups or sets—for example, genes in the same pathway or protein-protein interaction network[19–21] can be grouped into user-defined gene modules.

Lastly, *trans*-PCO tests for association between each SNP and the expression levels of the genes in each gene module by adapting the PCO method, which combines multiple gene expression PCs by using six PC-based statistical tests (STAR Methods). Each PC-based test combines multiple PCs uniquely, which allow signals under various genetic architectures to be captured. PCO evaluates the six PC-based tests and takes the minimum p value as the final test statistic. The final p values are computed according to Liu and Lin[15] (also see STAR Methods). Only PCs with eigenvalues $\lambda_k > 0.1$ are used in *trans*-PCO (Figure S1; Methods S1). To avoid identifying associations driven by *cis* effects, we excluded from the module all of the genes on the same chromosome as the test SNP. To correct for multiple testing, we performed 10 permutations to establish an empirical null distribution of p values (Methods S1).

### *Trans*-PCO outperforms existing methods in simulations

We performed simulations to evaluate the power of *trans*-PCO in detecting *trans*-eQTLs associated with multiple genes. We primarily compared the power to (1) the standard univariate test ("MinP") and (2) the PC1-based test (Kolberg et al.[11]; STAR Methods). We used a co-expression gene module consisting of 101 genes from the DGN dataset (module 29). In power simulations, we simulated a proportion of 101 genes in the module to be causal with nonzero effects generated from a point normal distribution (STAR Methods). We simulated the *trans* genetic variance to be 0.001, which is a low and realistic per SNP heritability for *trans* effects. In null simulations, all SNPs effects have the same *trans* genetic variance but zero average effects (STAR Methods).

*Trans*-PCO significantly outperformed the univariate test and the PC1 method across different sample sizes and proportions of causal genes (Figure 2). Specifically, the power of *trans*-PCO increases rapidly with increasing sample sizes. At a sample size of 800, assuming 30% of genes have causal effects in the

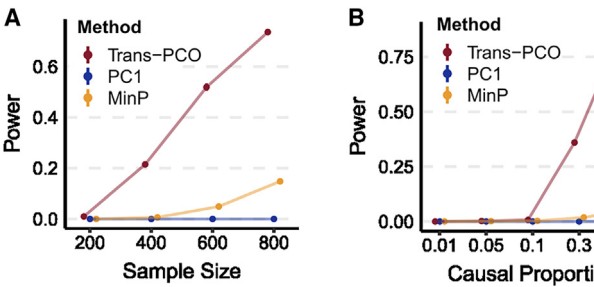

**Figure 2. Power of *trans*-PCO across different sample sizes and causal gene proportions, in comparison to PC1 and univariate (MinP) methods**

Points show average power across 1,000 simulations. Error bars representing 95% confidence intervals (CIs) are plotted, but many are too small to be visible. See numerical results in Table S2.

(A) Power comparison across various sample sizes. *Trans*-genetic variance was simulated to be 0.001, and the proportion of causal genes in the gene module was 30%.

(B) Power comparison across different proportions of causal genes in the gene module. The simulated sample size was 500.

gene module, the power of *trans*-PCO is 74%, compared to 15% for the univariate test and 0.0018% for the PC1 method (Figure 2A).

We also compared the power of each method across various causal gene proportions using a fixed sample size (500). All 3 methods have little power in detecting *trans*-eQTLs when the proportion of causal genes is below 10%. However, above this threshold, the power of *trans*-PCO increases dramatically: 36% at 30% causal genes and 86% at 50% causal genes. In contrast, the univariate and the PC1 methods remain almost powerless for nearly all of the simulated scenarios (Figure 2B). We note that the PC1 method appears to be almost powerless across the scenarios, which agrees with the previous observation that the PC1 can be less powerful than higher-order PCs in GWASs.[22] Simulation results at various genetic variances can be found in the supplemental information, including at extremely low proportions of causal genes and high *trans* effects (Figure S2). We found that the univariate method only outperforms *trans*-PCO when the proportion of causal genes is extremely low, such as only one causal gene in the entire gene set, and the *trans* effects are large. *Trans*-PCO gains more power when there are >1 causal gene because it aggregates multiple weak effects to improve power. Null simulations demonstrated that all three methods are well controlled for false positive inflations (Figure S3).

We included comparisons to two additional methods: ARCHIE, proposed by Dutta et al.,[12] and a method by Rotival et al.[10] (Figures S4 and S5; Methods S2). We showed that ARCHIE is not powerful at detecting *trans*-eQTL effects from a SNP to multiple genes, which are the effects for which *trans*-PCO was designed (Figure S4). We note that the main goal of ARCHIE is to identify trait-specific gene sets associated with GWAS loci, whereas *trans*-PCO is designed to map *trans*-eQTLs for any user-specified gene sets in specific tissues or cell types (discussion; Figure S4; Methods S2). The method of Rotival et al.[10] is based on the PC1-based approach, and we showed

that the method has limited power at identifying weak *trans*-eQTL effects (Figure S5; Methods S2).

## *Trans*-PCO identifies 3,899 *trans*-eSNP-module pairs associated with co-expression gene modules in the DGN dataset

We used *trans*-PCO to identify *trans*-eQTLs associated with co-expression gene modules in RNA-seq data from whole-blood samples of the DGN cohort (N = 913).[16] WGCNA[18] identified 166 co-expression gene modules, with the number of genes in each module ranging between 625 (module 1 [M1]) and 10 (M166) (Table S1). We then performed genome-wide scans of *trans*-eQTLs for each gene module. At a 10% false discovery rate (FDR), *trans*-PCO identified significant *trans*-eQTLs for 102 of 166 gene modules, corresponding to 3,899 significant *trans*-eSNP-module pairs (Table S3). Many *trans*-eSNPs are in linkage disequilibrium (LD). Using LD clumping to group *trans*-eSNPs into LD-independent loci ($R^2 < 0.2$), we found 202 *trans*-loci-module pairs (Figures 3A and S6; Tables S3 and S4).

We compared *trans*-eQTL signals detected in DGN by *trans*-PCO to signals identified by the univariate method in Battle et al.[16] Of the 12,132 genes analyzed by *trans*-PCO, the univariate method detected 326 significant *trans*-eSNP-gene pairs for 128 genes at 5% FDR.[16] At the same FDR level, *trans*-PCO identified 3,031 significant *trans*-eSNP-gene module pairs for 75 gene modules. We compared the magnitude of the significant *trans*-eQTL effects detected by *trans*-PCO and the univariate method. More specifically, we compared the maximum univariate $Z$ scores of SNPs and each gene in significant *trans*-eSNP-module pairs identified by *trans*-PCO to the $Z$ scores of significant *trans*-eSNP-gene pairs by the univariate method. We found that the maximum $Z$ scores of *trans*-PCO signals are much smaller than $Z$ scores of the univariate method signals (Figure 3B), indicating that our multivariate approach can detect much smaller *trans* effects than univariate methods.

We also applied the PC1 method (Kolberg et al.[11]) to DGN and identified 1,483 significant *trans*-eSNP-module pairs (55 *trans*-loci-module pairs) at 10% FDR, and 1,464 pairs (99%) were detected by *trans*-PCO (Figure S7A). Notably, in total, *trans*-PCO identified more than twice the signals of the PC1 method. However, the PC1 method identified more signals than expected because it was previously shown to be powerless in the simulations. We note that we simulated weak effects and sparse causal proportions to better reflect common and realistic *trans* effects, and the PC1 method is powerless in these settings. We performed additional simulations with large effects and high causal proportions, and effects with aligned direction of the PC1,[15] and the PC1 method achieved 50% power as *trans*-PCO or even the best power (Figures S8 and S9; Methods S2). In addition, we found in the DGN dataset that the univariate $Z$ scores of *trans* signals detected by the PC1 method are larger than those of *trans*-PCO signals (Figures S7B–S7D). Therefore, the *trans* signals detected by the PC1 method are likely of strong *trans* effects, and *trans*-PCO is able to detect additional weak *trans* effects.

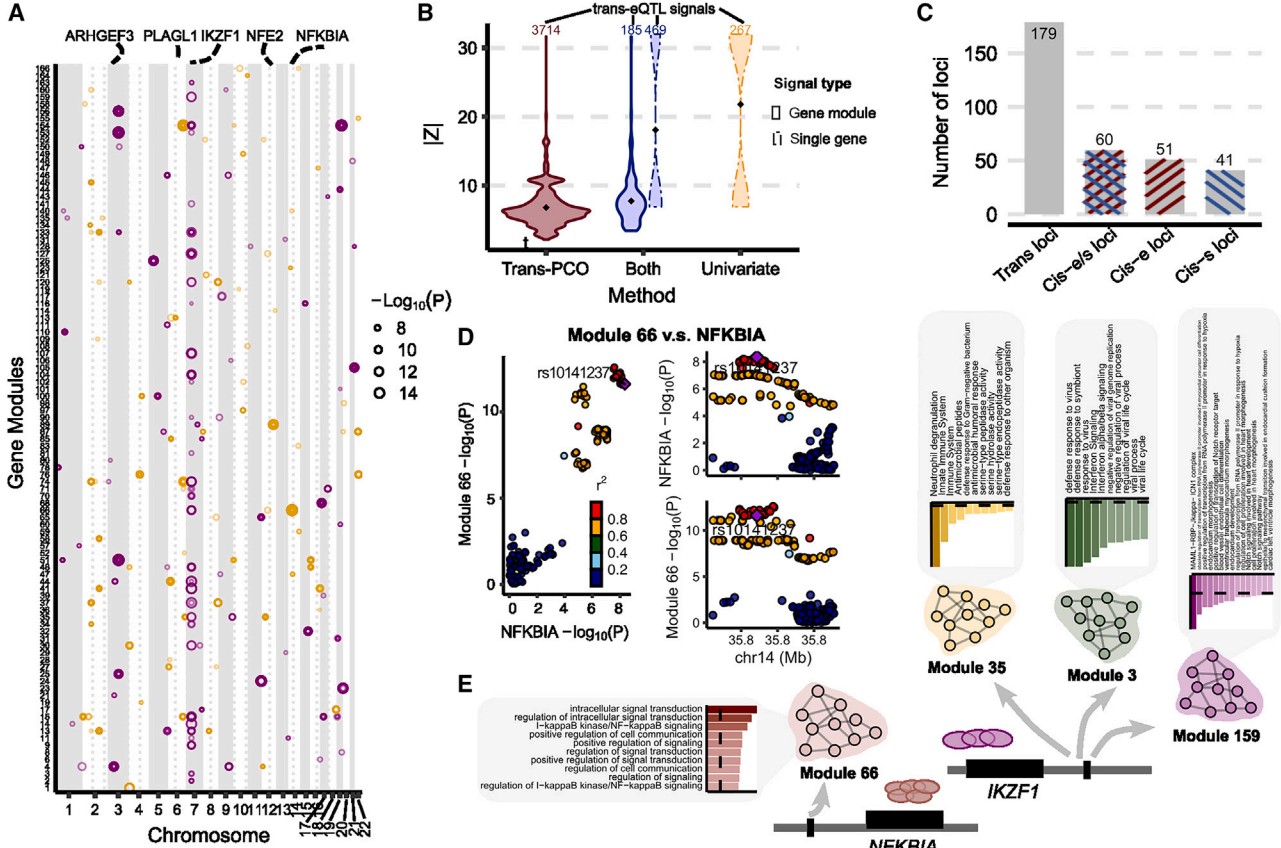

**Figure 3. *Trans*-PCO identifies *trans*-eQTLs associated with co-expression gene modules in DGN**

(A) Significant *trans*-eQTL signals associated with 166 co-expression modules in DGN. Chromosomal positions of *trans*-eSNPs are on the x axis, and gene modules are on the y axis. Point sizes are $-\log_{10}(p)$ values of significant *trans*-eQTLs. Purple and orange represent odd and even chromosomes, respectively.

(B) Comparison of the magnitude of significant *trans*-eQTLs effects detected by *trans*-PCO and the univariate method. The x axis shows signal categories: *trans*-PCO specific signals (*Trans*-PCO), univariate test specific signals (Univariate), and signals identified by both methods (Both). The maximum $Z$ scores of each SNP and each gene in a gene module is used to represent the SNP-module pair. The numbers on top are the number of signals in each category. Line type represents the target type of signals (gene module vs. single gene). The y axis is the absolute value of the $Z$ scores of the signals.

(C) Colocalization of *trans*-eQTLs and *cis*-e/sQTLs. The gray bar represents the *trans*-region used for colocalization analyses. The bar highlighted in blue represents the *trans*-region colocalized with *cis*-sQTLs, red for *cis*-eQTLs, and mixed color for either *cis*-eQTLs or *cis*-sQTLs.

(D) Colocalization of *trans*-eQTLs of M66 and *cis*-eQTLs of *NFKBIA*.

(E) Functional annotations of gene sets facilitate functional interpretation of *trans*-eQTL signals. The *trans*-eQTLs near *NFKBIA* and *IKZF1* are associated with several gene modules. The bar plots show the functional enrichments in modules. The numerical values of enrichments are in Table S7.

## *Trans*-eQTLs are enriched in variants with *cis*-regulatory effects on transcription factors

We found that only 31 *trans*-eSNPs (1%) are in coding regions, suggesting that a very small proportion of *trans*-eQTLs affect gene expression levels in *trans* by altering protein coding sequences. Several studies have shown that *trans*-eQTLs have *cis*-regulatory effects, affecting the expression levels or splicing of nearby genes[9,16]; thus, we evaluated our identified *trans*-eQTLs for concomitant *cis*-regulatory activity. We first overlapped *trans*-eSNPs with *cis*-eQTLs and *cis*-splicing QTLs (*cis*-sQTLs) in DGN.[23] Of the 2,955 *trans*-eSNPs (Table S3), we found that 71% are significant *cis*-eSNPs in DGN and 46% are significant *cis*-sSNPs, together accounting for 73% of all *trans*-eSNPs. To further examine whether the *cis* and *trans* effects are driven by the same variant, we performed colocalization analysis of *trans*-eQTLs with *cis*-eQTLs and *cis*-sQTLs using coloc[24]

(STAR Methods). Specifically, we first grouped *trans*-eSNP-gene module pairs into 179 *trans*-region-gene module pairs, based on 200-kb fixed-width regions (STAR Methods). We then performed colocalization analyses between the *trans*-eQTLs and *cis*-eQTLs/*cis*-sQTLs. We found that 51 of 179 *trans* regions colocalized with a *cis*-eQTL (the posterior probability of colocalized signals PP4 > 0.75; Figures 3C and S10). A total of 41 *trans*-regions colocalized with a *cis*-sQTL. Overall, 60 *trans*-regions shared causal variants with at least one *cis*-eQTL or *cis*-sQTL (Figure 3C; Table S5), confirming that *trans*-eQTL effects are generally mediated through *cis*-gene regulation. In addition, a large fraction of *trans*-loci (66%) do not colocalize with *cis*-eQTLs or *cis*-sQTLs. Although power may have limited our ability to detect colocalization of some *trans*-eQTLs and *cis*-eQTLs, there may also exist unknown *trans*-regulatory mechanisms, independent of *cis*-gene expression or splicing, which is subject to future studies.

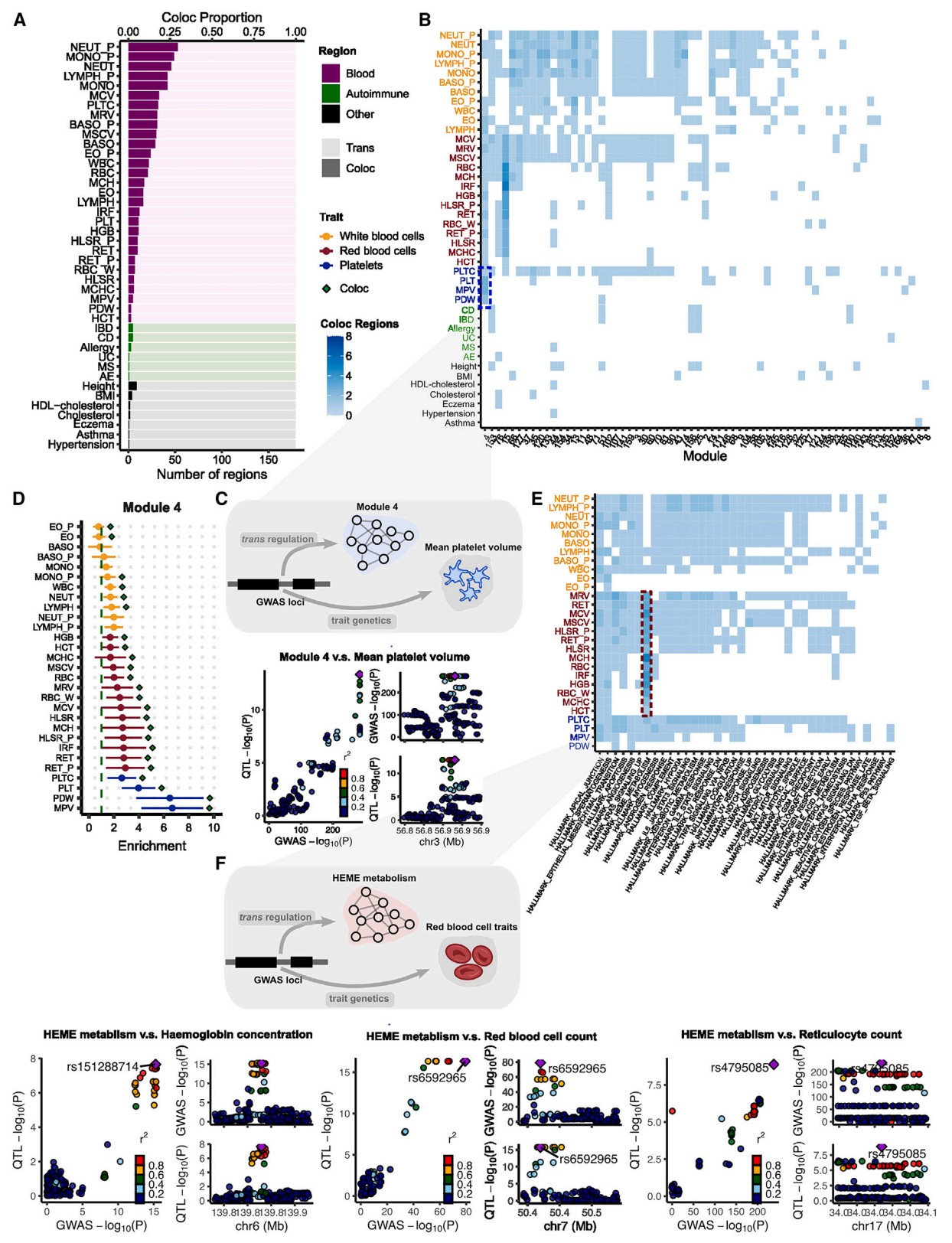

*(legend on next page)*

We also investigated the types and functions of genes that are likely to mediate *trans*-eQTL effects. We found that the genes nearest *trans*-eQTLs are highly enriched in "RNA polymerase II transcription regulatory region sequence-specific DNA binding" (adjusted p = $1.26 \times 10^{-3}$) and "DNA-binding transcription factor activity" (adjusted p = $1.39 \times 10^{-3}$; Table S6), suggesting that transcription factors are important mediators of *trans*-eQTL effects. Indeed, *trans*-PCO identified and replicated several well-known master *trans* regulators in blood, such as *IKZF1*,[17,25,26] *NFKBIA*,[17] *NFE2*,[9,17,27] and *PLAGL1*[17,26] (Figure 3A). We also found colocalization of these *trans*-eQTLs with *cis*-eQTLs at the *NFKBIA*, *NFE2*, and *PLAGL1* loci (Figures 3D and S10), supporting the conclusion that these genes are likely the *cis*-mediating genes.

### High-quality map of *trans*-eSNP to gene module associations improves functional interpretation

Most of the gene modules used in *trans*-PCO have functional annotations, which allows us to interpret the functional roles of the *trans*-eQTLs identified by the method. We first functionally annotated the 166 co-expression modules using g:Profiler,[28] which performs functional enrichment analysis on gene sets using predefined Gene Ontology and pathway annotations. This allowed us to annotate 131 of the 166 modules with at least 1 significantly enriched Gene Ontology or pathway (Table S7).

These annotations helped us interpret the function of identified *trans* effects. For example, the *trans*-eQTL signal near *IKZF1* (on chromosome 7) is significantly associated with 27 gene modules. *IKZF1* encodes a transcription factor, IKAROS, that belongs to the family of zinc finger DNA-binding proteins.[29] The *IKZF1* (IKAROS) *trans*-target gene M159 is significantly enriched in the "positive regulation of transcription of Notch receptor target" (adjusted p = $6.82 \times 10^{-3}$; Figure 3E). We were reassured to find that it previously had been found that IKAROS is a repressor of many Notch targets, and our *trans*-eQTL signal further supports the *trans* regulation of Notch signaling pathway by IKAROS.[30] *IKZF1 trans*-target M3 is significantly enriched in the Gene Ontology term "defense response to virus" (Figure S11; adjusted p = $8.7 \times 10^{-31}$), and M35 is significantly enriched in the innate immune system (adjusted p = $4.09 \times 10^{-17}$). These data support the conclusion that the *IKZF1* locus plays a *trans*-regulatory role in immune responses (Figure 3E). The *trans*-eQTLs near *NFKBIA*, which encode nuclear factor (NF)-κB inhibitor subunit A, are significantly associated with M66 (p < $1.8 \times 10^{-7}$). Interestingly, we found that M66 is highly enriched in NF-κB signaling pathway (adjusted p = $8.35 \times 10^{-5}$; Figure 3E), which supports the *trans*-regulation of the NF-κB signaling pathway by *NFKBIA*. The complete list of *trans*-eQTLs signals and func-

tional annotations of *trans*-target gene modules can be found in Tables S4 and S7.

### *Trans*-PCO identifies 965 *trans*-eSNP-module pairs associated with well-defined biological processes

To further demonstrate the utility of *trans*-PCO, we applied *trans*-PCO to 50 Human Molecular Signatures Database (MSigDB) hallmark gene sets, which represent well-defined biological processes,[19] including DNA repair, coagulation, heme metabolism, and Notch signaling (Table S8). Each gene set contains between 32 and 200 genes. In DGN, we identified 965 significant *trans*-eSNP-module pairs, corresponding to 41 gene sets and 120 *trans*-loci-module pairs ($R^2$ < 0.2), at a 10% FDR level (Figure S12; Tables S3 and S9).

*Trans*-eQTLs associated with well-defined biological processes facilitate the interpretation of the *trans*-eQTL signals. For example, we identified several *trans*-eQTL signals at the *NLRC5* locus (Table S9). The *trans*-target gene set is the "interferon alpha response" gene set, suggesting *trans* regulation from *NLRC5* to the interferon signaling pathway. Earlier studies have confirmed that *NLRC5* is a master regulator for major histocompatibility complex (MHC) class II genes and negatively regulates the interferon signaling pathway.[31,32] The *trans*-eQTL signals also validated our previous interpretations of *trans*-eQTLs associated with co-expression gene modules. For example, in agreement with our analysis of co-expression modules, we found that the *IKZF1* locus is significantly associated with several immune-related biological processes, such as interferon-gamma response (Figure 3E; Table S9).

### *Trans*-PCO improves understanding of *trans*-regulatory effects of disease-associated loci

To understand the *trans*-regulatory effects of genetic variants associated with complex traits, we performed colocalization analysis of *trans*-eQTL signals with GWAS loci of 46 complex traits and diseases, including 29 blood traits and 8 other common complex traits (e.g., height, body mass index) from the UK Biobank,[27,33] provided by Neale Lab (http://www.nealelab.is/uk-biobank/), and 9 autoimmune diseases[23,34–40] (Table S10; STAR Methods).

We grouped the *trans*-eSNPs into 200-kb regions (or *trans*-regions) for colocalization analyses (STAR Methods). The 3,899 *trans*-eQTLs associated with co-expression gene modules were grouped into 179 *trans*-region-module pairs. Of the 46 complex traits, 42 have at least 1 GWAS loci colocalized with 1 of 179 *trans*-region-module pairs. On average across all of the traits, 8.8% of *trans*-loci colocalize with GWAS loci

---

**Figure 4. Colocalization of *trans*-eQTLs with GWAS loci of 42 complex traits with at least 1 colocalization region**

(A) The number of colocalized *trans*-regions associated with co-expression gene modules with GWAS loci. The proportion of colocalization is the proportion of colocalized *trans*-loci over 179 *trans*-regions.

(B) Heatmap of the number of colocalized *trans*-regions associated with co-expression gene modules with GWAS loci between each module and trait. Tiles represent the number of colocalized regions.

(C) Colocalization of mean platelet volume-associated locus near *ARHGEF3* and *trans*-eQTL of M4.

(D) Heritability enrichment of M4 in blood traits. Error bars are 95% CIs.

(E) Heatmap of the number of colocalized *trans*-regions associated with MSigDB hallmark gene sets with GWAS loci.

(F) Colocalization of GWAS loci-associated red blood cell traits and *trans*-eQTLs associated with heme metabolism. Six loci associated with red blood cell traits are associated with heme metabolism in *trans*. Numerical results can be found in Table S14. Colocalization plots of the other loci are in Figure S13.

(Figure 4A; Table S11). We observed a higher proportion of co-localization with blood traits (mean proportion 12.0%) than non-blood traits (mean proportion 1.5%). Although we expect some higher proportions of colocalization with blood traits to occur in a whole-blood sample, our results may also indicate some residual effects due to cell composition, despite corrections for cell composition using both gene expression PCs and estimated cell-type proportions,[16] such that some *trans*-eQTLs may regulate the abundance of cell proportions and therefore are associated genes that are specifically expressed in certain cell types (discussion). Our results are consistent with a recent study by the eQTLGen consortium, which has shown that *trans*-eQTLs in whole blood reflect a combination of cell-type composition and intracellular effects.[9]

Nevertheless, we found several *trans*-eQTLs that colocalized with GWAS loci, which revealed specific interpretable pathways or functional gene sets (Figure 4B; Table S12). For example, *trans*-eQTLs associated with co-expression M4 colocalized with 24 of 29 blood traits (Figure 4B). M4 is highly enriched for genes involved in platelet activation (adjusted p = 1.12 × $10^{-12}$; Figure S11; Table S7). One of the colocalized *trans*-eSNPs associated with M4 is in the introns of the *ARHGEF3* gene (Figure 4C), which has been shown to play a significant role in platelet size in mice.[41] To further support the interpretation of colocalized signals, we estimated heritability enrichment of M4 in blood traits using stratified LDscore regression[42] (S-LDSC; Figures 4D and S13). We reasoned that an enrichment of trait heritability near genes in a module would strongly support the involvement of a module in the genetic etiology of a trait. Strikingly, we found that M4 is significantly enriched in the heritability of multiple blood traits, and that the enrichment was especially strong for platelet traits such as platelet distribution width (odds ratio [OR] = 6.5, p = 7.0 × $10^{-5}$) and mean platelet volume (OR = 6.7, p = 1.2 × $10^{-5}$; Figure 4D; Table S13). In addition, we evaluated whether M4 genes are significantly enriched in genes associated with platelet traits, identified by transcriptome-wide association studies (TWASs). There are 1,339 unique genes significantly associated with platelet traits in the UK Biobank.[43] M4 genes are significantly enriched in TWAS genes associated with platelet traits (88 overlap genes, p = 6.7 × $10^{-10}$, Fisher's exact test), which further supports the role of M4 in platelet traits. Finally, we identified that the *ARHGEF3* locus is significantly associated with the MSigDB coagulation hallmark gene set (Table S9). These findings strengthen the model in which genetic variation near *ARHGEF3* affects the expression levels of multiple genes that are involved in platelet biology and that also harbor nearby genetic variation associated with platelet traits.

We also performed colocalization analysis of *trans*-eQTLs-associated MSigDB hallmark gene sets (Figure 4E; Table S14). One of the gene sets represents heme metabolism, which is an essential process underlying erythroblast differentiation and red blood cell counts. We found that six *trans*-eQTL loci of heme metabolism significantly colocalized with GWAS loci associated with red blood cell traits, such as hemoglobin concentration, red blood cell count, and reticulocyte count (PP4 = 0.76–1.00; Figures 4F and S14; Table S14). We found that the genes in the gene sets are significantly enriched in TWAS-significant genes associated with hemoglobin levels in the UK Biobank (35 overlap genes, p = 8.1 × $10^{-4}$, Fisher's exact test), which further supports the role of the hallmark gene set in red blood cell traits. Our results provide evidence that these six loci regulate heme metabolism in *trans*, which is an essential process underlying erythroblast differentiation and red blood cell counts.

In another example, we found a *trans*-eQTL near *IKZF1* for M3 that colocalizes with 11 blood traits, 7 of which are related to white blood cells (Table S12). As mentioned previously, M3 is significantly enriched for Gene Ontology terms, including "defense response to virus" (adjusted p = 8.7 × $10^{-31}$) and "negative regulation of viral processes" (adjusted p = 1.07 × $10^{-17}$; Table S7). The enrichments are driven by many genes related to interferon (e.g., *IFI6*, *IFI16*, *IRF7*), which are proteins released by host cells in response to the presence of viruses and indicate immune-related functions (Tables S1 and S7). In addition, our heritability analysis of genes in M3 identified enrichments for multiple traits associated with blood cell-type count, including neutrophil count (OR = 2.3, p = 1.7 × $10^{-4}$) and white blood cell count (OR = 2.1, p = 1.3 × $10^{-4}$, Figure S15). Our analyses support that the white blood cell associated locus *IKZF1* regulates immune-response pathways in *trans*.

Taken together, our functional map of *trans*-eQTLs revealed concrete examples where genetic variants associated with complex traits also influence a biological pathway or a coherent set of genes with similar functions. Thus, *trans*-eQTL of gene sets have the potential to reveal *trans*-regulatory mechanisms underlying complex traits and diseases. The complete list of colocalization signals for each trait can be found in Table S12.

## Summary statistics-based *trans*-PCO identified 10,167 *trans*-eSNP-module pairs in eQTLGen

We developed summary statistics-based *trans*-PCO to increase its applicability to gene expression datasets of large sample sizes, such as eQTLGen[9] (N = 31,684, whole blood). To ensure that summary statistics-based *trans*-PCO signals are well controlled for test statistics inflation and false positives, we added two steps to the original pipeline. First, we carefully selected gene sets to minimize the noise when approximating the gene correlation matrices. When only summary statistics are available, the correlation matrix of each gene set is approximated with the correlations of Z scores of the insignificantly associated SNPs of each gene. A low ratio of SNPs to genes (<50) results in a noisy approximation of correlation matrices and test statistics inflation (Figure 5A; STAR Methods; Methods S3). Therefore, we only used gene modules with ratios >50 to test for *trans*-eQTLs, which we show are well controlled for inflation (Figures 5A and S16). Second, we removed genes in the module that were cross-mappable to the test SNP loci (STAR Methods) in the association test to reduce false positives caused by multimapping reads.

The eQTLGen study performed the standard univariate *trans*-eQTL mapping on a subset of 10,317 GWAS SNPs, and the summary statistics of these *trans*-eQTLs are available. We applied the summary statistics-based *trans*-PCO to these summary statistics to identify *trans*-eQTLs-associated co-expression gene modules and MSigDB hallmark gene sets.

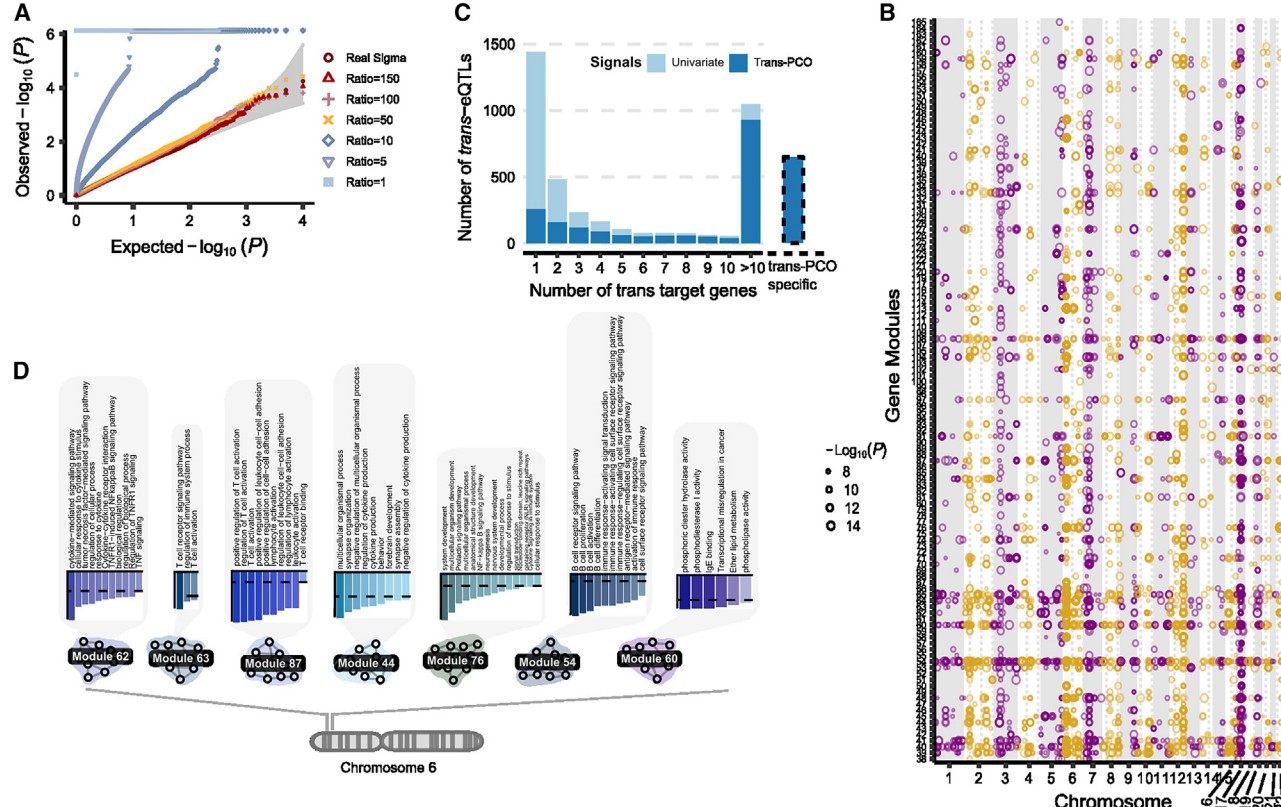

**Figure 5. Trans-PCO identifies trans-eQTLs associated with co-expression gene modules and MSigDB hallmark gene sets in eQTLGen**

(A) Summary statistics-based trans-PCO is well controlled for test statistics inflations. We show gene module 1 (size 625) as an example. SNP-to-gene ratios used for correlation matrix estimation are in different shapes and colors. Red-yellow shades represent higher ratios ($\geq 50$), and blue shades represent lower ratios. Gray area shows 95% CIs. Trans-PCO used a minimum ratio of 50.

(B) 8,199 significant trans-eSNP-module pairs associated with co-expression modules in eQTLGen. Chromosomal positions of trans-eSNPs are on the x axis and gene modules are on the y axis. Point sizes are $-\log_{10}(p)$ values of significant trans-eQTLs.

(C) The majority of hub SNPs targeting >10 genes in the original eQTLGen study are identified by trans-PCO. The light blue bar represents the total number of trans-eQTLs in the original eQTLGen study at 5% FDR level. The dark blue bar represents the trans-eQTLs also detected by trans-PCO under Bonferroni correction that are associated with co-expression modules or MSigDB gene sets. The bar at right shows the trans-eQTLs detected only by trans-PCO.

(D) The HLA locus is associated with several immune-related gene modules in trans. The bar plots show the functional enrichment of co-expression gene modules.

Of the 166 co-expression gene modules identified in DGN, we used 129 modules with reliable correlation matrix approximations to ensure that the trans-eQTL signals were well controlled for inflation (Figures 5A and S16; STAR Methods; Methods S3). Similarly, of the 50 MSigDB hallmark gene sets, we only used 11 gene sets with accurate correlation matrix approximations (Figure S17). In total, there were 4,533 genes in the tested co-expression gene modules and hallmark gene sets. For co-expression gene modules, we identified 8,116 trans-eSNP-gene co-expression module pairs, corresponding to 2,161 eQTLGen test SNPs and 122 gene modules (Figure 5B; Tables S3 and S15). For hallmark gene sets, we found 2,051 significant trans-eSNP-hallmark gene set pairs, corresponding to 1,018 SNPs and all 11 hallmark gene sets, using Bonferroni correction (Tables S3 and S16). In eQTLGen, we did not perform LD clumping on trans-eSNPs because they were GWAS SNPs associated with different traits and diseases. The univariate method used in eQTLGen[9] identified 1,050 hub SNPs target-ing >10 genes at 5% FDR, 89% of which were also identified by trans-PCO (Figure 5C).

The large sample size in eQTLGen improves the power of trans-eQTL detection. Of the 3,899 significant trans-eSNP-co-expression module pairs in DGN, 38 pairs were also tested in eQTLGen. We did find that all 38 trans signals were replicated in eQTLGen (under a replication p value cutoff of 0.1/38; Table S17) and all association p values were highly significant ($p < 10^{-12}$; Figure S18). In contrast, most of the trans-eQTL signals in eQTLGen were not found in DGN. For example, of the 7,577 SNP-module pairs analyzed in both datasets, there were 7,291 pairs (96%) that were uniquely identified in eQTLGen (defined as at least 1 MB away from trans-eQTL SNPs in DGN). This is not surprising because the association p values are much smaller in the eQTLGen dataset due to the larger sample size (Figure S19). Similarly, 8 significant trans-eSNP–hallmark gene set pairs in DGN were tested in eQTLGen, and all of them were replicated. We also compared eQTLGen signals by

*trans*-PCO to those identified by ARCHIE in Dutta et al.[12] (Figure S4; Methods S2).

The nearest genes of eQTLGen *trans*-eQTLs are significantly enriched in DNA-binding activity (adjusted p = $3.73 \times 10^{-4}$) and transcription factor binding (adjusted p = $1.74 \times 10^{-7}$), as well as immune responses such as cytokine receptor activity (adjusted p = $7.27 \times 10^{-7}$) or MHC class II receptor activity (adjusted p = $9.93 \times 10^{-5}$; Figure 5B; Table S18). We found that the enrichment of immune responses was driven by *trans*-eQTLs in the human leukocyte antigen (HLA) region on chromosome 6 (e.g., *HLA-DRA*, *HLA-DRB1*; Table S15) or near cytokine receptor genes (e.g., *IL23R*, *IL1R1*, *CXCR4*; genes on the chemokine receptor gene cluster region: *CCR2*, *CCR3*, *CCR5*, etc.). These *trans*-eQTLs are associated with several autoimmune diseases, such as type 1 diabetes, autoimmune thyroid diseases, cutaneous lupus erythematosus, and inflammatory bowel disease (Table S15). The *trans*-PCO signals help us understand the *trans*-regulatory mechanism of these loci. For example, we found that the *trans*-target gene modules of the HLA loci are enriched in immune-related functions, such as cytokine production (M44), B cell differentiation (M54), immunoglobulin E (IgE) binding (M60), tumor necrosis factor signaling pathway (M62), T cell activation (M63 and M87), and cytokine signaling pathway (M62 and M76; Figure 5D). The *IL23R* locus is associated with cytokine signaling pathway (M76) in *trans*. The chemokine receptor genes were associated with several gene modules, including cytokine production (M44), IgE binding (M60), and T cell activation (M87). These *trans*-eQTL signals support the conclusion that genetic loci associated with autoimmune disease regulate immune-related pathways in *trans*.

## DISCUSSION

In summary, we developed a powerful method, *trans*-PCO, to detect *trans*-eQTLs associated with expression levels of co-expressed or co-regulated genes. The multivariate approach of *trans*-PCO can detect much smaller *trans* effects and is substantially more powerful than existing methods.

We thoroughly compared the performance of *trans*-PCO versus other methods, such as the PC1-based method by Kolberg et al.,[11] ARCHIE by Dutta et al.,[12] and Rotival et al.[10] (Figures 2, S4, S5, and S7–S9; Methods S2). *Trans*-PCO- and the PC1-based method are both designed to identify individual *trans*-eQTLs of any gene sets containing multiple genes, and the comparison between them is straightforward. However, ARCHIE is different and not directly comparable to the other two methods for several reasons (see more discussions in Methods S2). First, ARCHIE captures only trait-specific *trans*-regulations, by testing significance against a null hypothesis based on a subset of genetic variants that are trait associated. In contrast, *trans*-PCO identifies *trans*-eQTLs under the general null hypothesis with no additional assumptions. Second, *trans*-PCO and ARCHIE are designed to capture different *trans*-regulatory effects. ARCHIE is powerful when multiple disease-associated variants have weak effects on a single gene or multiple disease-associated variants have weak effects on multiple genes (Figure 2 in Dutta et al.[12]), which are not co-regulated by a shared *trans* genetic locus. In contrast, *trans*-PCO is designed to capture weak *trans* signals of a variant on multiple co-regulated genes (Figure S4; Methods S2). Third, ARCHIE identifies components consisting of multiple trait-associated SNPs and multiple genes, without knowing the exact *trans*-eQTL SNP driving the *trans*-regulation. It is hard to further study *trans*-regulatory mechanisms of the *trans*-eQTLs. Fourth, ARCHIE takes all of the genes as input and infers gene sets that are *trans*-regulated by disease-associated variants, whereas *trans*-PCO is flexible to be applied to any user-defined gene set of interest to identify *trans*-eQTLs. In summary, *trans*-PCO and ARCHIE have different goals and are designed for detecting different types of *trans* signals. However, we thoroughly compared ARCHIE and *trans*-PCO in both simulations and real data analyses (Methods S2). We believe that these comparisons will provide insights into when and how these methods should best be used.

*Trans*-eQTLs identified in bulk tissues can be a combination of cell composition *trans*-eQTLs, which are driven by cell-type proportions, and intracellular *trans*-eQTLs, which capture *trans*-regulatory effects in a single cell type. In our analysis of DGN dataset, we included the estimated cell proportions as covariates, in addition to gene expression PCs, to obtain higher proportions of intracellular *trans*-eQTLs. Co-expression gene modules could also capture cell proportion effects. In our study, we removed cell proportions from gene expression levels before clustering genes into co-expression modules. Although this can correct for cell proportion effects in the co-expression modules to some extent, we note that it does not guarantee their complete removal.

Many studies, including ours, seek to avoid cell composition effects. However, by closely examining *trans*-eQTLs discovered in our study, we think that cell composition *trans*-eQTLs can also be biologically interesting. For example, the *IKZF1* locus is significantly associated with several gene modules enriched with viral defense and other immune-related functions in *trans*. The locus is also significantly associated with white blood cell proportions. Given the general function of white blood cells in fighting infections, these observations raise the possibility that the *trans*-eQTLs near *IKZF1* regulate antiviral activity by affecting white blood cell-type proportion. Supporting this hypothesis, we found earlier that genetic variants near *IKZF1* are also associated with expression levels of genes in M159, which are enriched in genes involved in the Notch signaling pathway. The Notch signaling pathway plays a central role in cell proliferation, cell fate, and cell differentiation[44]; thus, our analyses reveal a plausible mode of action whereby genetic variants near *IKZF1* affect multiple immune-related functions by influencing white blood cell-type proportions.

Identifying the network effects of genetic variants not only shed light on molecular mechanisms of complex associated loci, but it can also have important translational applications. First, genes that are associated with disease-relevant pathways can serve as evidence for therapeutic targets of the disease. In a preliminary analysis, we examined whether allergy drug targets are more likely to be associated with immune-related gene sets. Among a total of 142 gene sets used for *trans*-eQTL identification in eQTLGen, 19 were defined as immune related. We used 55 launched allergy drug target genes from the Broad Institute Drug Repurposing Hub

(https://repo-hub.broadinstitute.org/repurposing), 5 of which are near allergy-associated loci in eQTLGen. Interestingly, we found all 5 targets to be associated with immune-related gene sets (Table S19). Detailed analyses can be found in STAR Methods and Methods S4. Although the enrichment is not statistically significant (p = 0.12, Fisher's exact test; Table S20), it is likely due to the small number of drug targets included in our analyses. In addition, we observed that the *trans*-gene modules of drug targets converge to gene sets whose functions are highly relevant to allergy. For example, three drug targets (*IL3*, *UGT3A1*, and *SLC37A4*) are associated with gene sets enriched for the B cell signaling pathway. Second, network effects of disease variants can be used for repurposing existing drug compounds to new diseases. Drug repurposing can substantially reduce cost and time to develop new treatments. If the gene expression profile of an existing drug is enriched for genes in the *trans*-network of associated loci of another disease, it can serve as evidence for repurposing. We believe comprehensive catalogs of *trans*-networks effects in human cell types and tissues will serve as important resources for the interpretation of *trans*-regulatory effects of disease-associated loci as well as translation applications. Therefore, we made all of the *trans*-PCO *trans*-eQTL signals, with functional annotation of the gene sets, publicly available, downloadable, and browsable in http://www.networks-liulab.org/transPCO.

### Limitations of the study

A limitation of multivariate association tests, including *trans*-PCO, is that they do not explicitly identify which genes in the gene sets are significantly associated with the test SNP. Although functional annotations of gene sets facilitate our understanding of the *trans*-eQTL signals, it is possible that the genes driving *trans* associations are different from the genes driving functional enrichment of the gene sets. Therefore, the biological interpretation of *trans*-eQTL signals should be supported with other evidence before it is considered definitive. However, there are exploratory analyses that can help prioritize genes in the network that are key drivers of the underlying signal. For example, by examining the univariate association p values between the *trans*-eQTL SNP and each gene in the network, the user can prioritize genes with the most significant p values as likely *trans*-targets. Furthermore, the users can also use the $\pi 1$ statistics on the univariate p values to estimate the proportion of genes that have true *trans* effects in the network. Although the exact molecular mechanism requires further validation, the large number of *trans*-eQTLs identified by *trans*-PCO in our study opens up new opportunities to understand complex traits-associated loci and underlying mechanisms.

### STAR★METHODS

Detailed methods are provided in the online version of this paper and include the following:

- KEY RESOURCES TABLE
- RESOURCE AVAILABILITY
  - ○ Lead contact
  - ○ Materials availability
  - ○ Data and code availability
- METHOD DETAILS
  - ○ *Trans*-PCO pipeline
  - ○ Simulation
  - ○ Genotype QC of DGN dataset
  - ○ Summary-statistics-based *trans*-PCO
- QUANTIFICATION AND STATISTICAL ANALYSIS
  - ○ Colocalization of *trans*-eQTLs and GWAS loci
  - ○ Colocalization of *trans*-eQTLs and *cis*-e/sQTLs
  - ○ Trait heritability enrichment in gene modules
  - ○ Association of drug targets with disease-relevant gene sets regulated in *trans*

### SUPPLEMENTAL INFORMATION

### ACKNOWLEDGMENTS

We thank Y. Li, A. Dahl, Y. Gilad, and Z. Mu for helpful discussions. We thank N. Gonzales, C. Jones, and S. Sumner for editing the manuscript. This work was completed in part with resources provided by the University of Chicago's Research Computing Center. This research was funded by the NIGMS Maximizing Investigators' Research Award (R35GM138084).

### AUTHOR CONTRIBUTIONS

L.W., Z.L., and X.L. developed the method. L.W. and X.L. designed all of the analyses. L.W. implemented the method and performed all of the data analyses under the supervision of X.L. L.W. and X.L. wrote the manuscript with input from all of the coauthors. N.B. created the website to share the results.

### DECLARATION OF INTERESTS

The authors declare no competing interests.

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

## STAR★METHODS

### KEY RESOURCES TABLE

| REAGENT or RESOURCE | SOURCE | IDENTIFIER |
| --- | --- | --- |
| **Deposited data** | | |
| Data generated in this paper, including all *trans*-eQTL signals, with functional annotation of the gene sets | This paper | http://www.networks-liulab.org/transPCO Zenodo link: https://zenodo.org/doi/10.5281/zenodo.10602699 |
| Depression Genes and Networks study (DGN) | Battle et al.[16] | Downloaded by application through the NIMH Center for Collaborative Genomic Studies on Mental Disorders, under the "Depression Genes and Networks study (D. Levinson, PI)." |
| eQTLGen summary statistics | Võsa et al.[9] | https://www.eqtlgen.org/ |
| MSigDB hallmark gene sets | Liberzon et al.[19] | http://www.gsea-msigdb.org/gsea/msigdb/human/genesets.jsp?collection=H |
| UK Biobank GWAS summary statistics | http://www.nealelab.is/uk-biobank/ | http://www.nealelab.is/uk-biobank/ |
| The ENCODE 36-mer of the reference human genome | N/A | https://hgdownload.soe.ucsc.edu/goldenPath/hg19/encodeDCC/wgEncodeMapability/wgEncodeCrgMapabilityAlign36mer.bigWig |
| **Software and algorithms** | | |
| All original code, related to the *trans*-PCO pipeline and code to reproduce analyses presented in this work | This paper | https://github.com/liliw-w/Trans Zenodo link: https://zenodo.org/doi/10.5281/zenodo.10602558 |
| TensorQTL | Taylor-Weiner et al.[45] | https://github.com/broadinstitute/tensorqtl |
| WGCNA | Langfelder et al.[18] | https://cran.r-project.org/web/packages/WGCNA/index.html |
| coloc | Giambartolomei et al.[24] | https://github.com/chr1swallace/coloc |
| S-LDSC | Finucane et al.[42] | https://github.com/bulik/ldsc |

## RESOURCE AVAILABILITY

### Lead contact
Further information and requests for resources and reagents should be directed to and will be fulfilled by the lead contact, Xuanyao Liu (xuanyao@uchicago.edu).

### Materials availability
No materials were generated in the study.

### Data and code availability
- All *trans*-eQTL signals, with functional annotation of the gene sets, can be browsed and downloaded at http://www.networks-liulab.org/transPCO, and has been deposited at Zenodo (https://zenodo.org/doi/10.5281/zenodo.10602699). Any additional data reported in this paper will be shared by the lead contact upon request. DOIs are listed in the key resources table.
- All original code, related to the *trans*-PCO pipeline and code to reproduce analyses presented in this work are publicly available at https://github.com/liliw-w/Trans, and has been deposited at Zenodo (https://zenodo.org/doi/10.5281/zenodo.10602558). DOIs are listed in the key resources table.
- Any additional information required to reanalyze the data reported in this paper is available from the lead contact upon request.

## METHOD DETAILS

### *Trans*-PCO pipeline
#### Data processing
*Trans*-PCO removes all reads that are mapped to low mappability regions, in addition to multi-mapped reads marked by alignment tools before quantifying gene expression levels. More specifically, we downloaded the mappability of 36-mer of the reference human

genome computed by the ENCODE project and defined genomic regions with a mappability score <1 (i.e., 36-mers that could be mapped to two or more different genomic regions) as low mappability regions. We removed reads mapped to low mappability regions allowing 2 mismatches.

Following thorough read removal, *trans*-PCO quantifies gene expression levels as Transcript Per Million (TPM). Gene expression levels were first quantile normalized across samples, and then normalized to a standard normal across genes. We also filtered out genes that are not protein-coding or lincRNA genes. Finally, to control for potential confounding factors and capture the co-expressed gene modules mainly driven by genetic effects, we regressed out covariates from the expression profiles. The typical covariates may include biological and technical covariates, such as genotype PCs, expression PCs, and blood cell type proportions etc.[16,17]

### Identification of gene co-expression networks

By default, *trans*-PCO uses WGCNA[18] to construct gene co-expression modules, where genes are connected through correlations among their residualized expression levels. WGCNA uses hierarchical clustering to cut the network into separate gene modules with highly correlated expression levels. We used the default parameter settings, except that we specified the minimum module size parameter ('minModuleSize') to 10 to obtain small gene modules. *Trans*-PCO also takes other pre-defined gene sets, such as genes in the same pathway or biological processes.

### Multivariate association test

We test if a genetic variant is associated with genes in a module through *trans* regulations using the multivariate model as follows,

$$[y_1 \cdots y_K] = G[\beta_1 \cdots \beta_K] + covariates + e$$

where $G$ is the dosage of a reference allele representing the genotype of an SNP, $\beta_k$ is the effect of the SNP on $k$-th gene in the module with $K$ genes, and $y_k$ is the expression level of the $k$-th gene. To test if an SNP of interest is significantly associated with the module, we test the null hypothesis,

$$H_0 : \beta_1 = \cdots = \beta_K = 0$$

We use a PC-based omnibus test (PCO),[15] which is a powerful and robust PC-based approach aiming at testing genetic association with multiple genes with no prior knowledge of the true effects.

Specifically, PCO combines multiple single PC-based tests in linear and non-linear ways, corresponding to a range of causal relationships between the genetic variant and genes, to achieve higher power and better robustness. A single PC-based test (most commonly the first primary $PC_1$) is,

$$T_{PC_k} = \mu_k^T Z \sim N(\mu_k^T \beta, \lambda_k), 1 \le k \le K$$

where $Z$ is a $K \times 1$ vector of univariate summary statistic $Z$ scores of the SNP for $K$ genes in a module, $\mu_k$ is the $k$-th eigenvector of the covariance matrix $\Sigma_{K \times K}$ of $Z$, $\lambda_k$ is the corresponding eigenvalue, and $\beta$ represents the true causal effect. PCO combines six PC-based tests, including,

$$PCMinP = min_{1 \le k \le K} p_k, and \ PCFisher = -2 \sum_{k=1}^{K} log(p_k),$$

where $p_k$ is the p value of $T_{PC_k}$. These two tests take the best p value of single PC-based tests and combine multiple PC p values as the test statistic. Other tests include,

$$PCLC = \sum_{k=1}^{K} \frac{T_{PC_k}}{\lambda_k}, WI = \sum_{k=1}^{K} T_{PC_k}^2, Wald = \sum_{k=1}^{K} \frac{T_{PC_k}^2}{\lambda_k}, VC = \sum_{k=1}^{K} \frac{T_{PC_k}^2}{\lambda_k^2}$$

which are linear and quadratic combinations of each single PC-based test weighted by eigenvalues. The six tests achieve best power in specific genetic settings with different true causal effects.[15] PCO takes the best p value of the PC-based tests as the final test statistic,

$$T_{PCO} = min \ p_{\{PCMinP, PCFisher, PCLC, WI, Wald, VC\}}$$

to achieve robustness under unknown genetic architectures while maintaining a high power. The p value of PCO test statistics can be computed by performing an inverse-normal transformation of the test statistics,

$$p_{T_{PCO}} = 1 - P\{min \ \Phi^{-1}(p_{\{PCMinP, PCFisher, PCLC, WI, Wald, VC\}}) > \Phi^{-1}(T_{PCO}^{obs})\}$$

where $\Phi^{-1}$ denotes the inverse standard normal cumulative distribution function. The p value can be efficiently computed using a multivariate normal distribution as described in Liu et al.[15]

To prevent *cis*-regulatory effects from driving the identified *trans* associations between an SNP and module, we removed genes in the module that are on the same chromosome as the tested variant. In addition, to avoid false positive signals in *trans* associations

due to alignment errors, we discarded RNA-seq reads that are mapped to multiple locations or poorly mapped genomic regions (mappability score <1)[16,17] before quantifying gene expression levels. We calculated the summary statistic $Z$ scores using TensorQTL.[45]

### Simulation

We performed simulations to evaluate power and type I error of *trans*-PCO, a univariate test ("MinP") and a primary PC-based test ("PC1"). The PC1 based test takes only the first PC as the proxy of a gene module and uses it as the response variable to test for genetic variants with significant associations. The MinP method uses the minimum p value across genes in the module to represent the association of the gene module. More specifically, the test statistics of the PC1 method is $T_{PC_1} = \mu_1^T Z \sim N(\mu_1^T \beta, \lambda_1)$, where $Z$ is the vector of z scores between the SNPs and each individual gene, $\mu_1$ is the first eigenvector of the covariance matrix $\Sigma_K$ of the $K$ genes, $\lambda_1$ is the corresponding eigenvalue, and $\beta$ represents the true causal effect. The p value of PC1 test statistics is computed based on $N(\mu_1^T \beta, \lambda_1)$. The test statistics of the MinP method is $T_{MinP} = min\{p_1^g, \cdots, p_K^g\}$, where $p_i^g$ is the association p value of gene i. The p value of MinP test statistics is $P_{MinP} = T_{MinP} \times K$, which uses Bonferroni correction.

We used a real gene module containing 101 genes (Module 29) from the DGN dataset in our simulations. The correlation matrix of the 101 genes is $\Sigma_{101}$. In null simulations, we simulated $Z$ scores of $10^7$ SNPs from the null distribution, $Z_{NULL} \sim N(0, \Sigma_{101})$. We applied the three methods to the simulated $Z$ scores and evaluated the p values to validate if the statistical tests are well calibrated for type I error.

In power simulations, we simulated 10k $Z$ scores of SNPs from the alternative distribution,

$$Z_{Alt} \sim N\left(\sqrt{n}\left[\beta_{101\gamma}, 0\right]^T, \Sigma_{101}\right)$$

where $n$ is the sample size, $\beta$ is a $101\gamma$-long vector representing the causal effect of an SNP on 101 genes, and $\gamma$ is the proportion of true target genes in the module with non-zero effects. We generated $\beta_k$ from a point normal distribution, where $\beta_k \sim N(0, \sigma_b^2)$ for proportion $\gamma$, and $\beta_k = 0$, otherwise. The *trans*-genetic variance is $\sigma_b^2$, which is a low and realistic per SNP heritability for *trans* effects. By default, we set the sample size $n$ to be 500, 30% genes (30) in the module are true *trans* target genes, and $\sigma_b^2$ to be 0.001.

To evaluate how three tests perform across different genetic architectures, we simulated multiple scenarios across varying sample sizes, target gene proportions, and genetic variances. Specifically, we looked at the cases where sample size is 200, 400, 600, and 800, causal genes proportion is 1%, 5%, 10%, 30%, and 50%, and genetic variance is 0.002, 0.003, 0.004, 0.005, and 0.006. We simulated 10k SNPs and performed 1000 simulations. To control the false discovery rate, we corrected the p values for multiple testing based on the simulated empirical null distribution of p values, to keep it consistent with the method used in the RNA-seq dataset (Methods S1). We set significance levels at 10% FDR to be consistent with real data analysis. We computed power as the average proportion of significant tests out of 10,000 simulated SNPs across 1000 simulations. An association is significant if its adjusted p value is lower than 0.1. We computed power as the average proportion of significant tests out of 10,000 simulated SNPs across 1000 simulations.

### Genotype QC of DGN dataset

We analyzed an RNA-seq dataset from whole blood.[16] We performed a series of QC on individuals, genotypes, RNA-seq reads, and genes before quantifying gene expression profiles. The QC of RNA-seq data and quantifying gene expression is included in the pre-processing steps of *trans*-PCO (see above). For individual-level QC, we removed related individuals from 922 samples and kept 913 individuals in total for further analysis. For genotype-level QC, we used SNPs with genotyping rate >99%, minor allele frequency >5%, and Hardy-Weinberg equilibrium $< 10^{-6}$. The detailed procedures were described in Liu et al.[17]

### Summary-statistics-based *trans*-PCO

The eQTLGen Consortium[9] has conducted the largest *cis*- and *trans*-eQTLs association analyses in blood to date. Specifically, 31,684 samples were tested for over 11 million SNPs across 37 cohorts. The summary statistics of *trans*-eQTLs are available for 10,317 trait-associated SNPs on 19,942 genes.

We applied our pipeline *trans*-PCO to eQTLGen summary statistics, using the same 166 co-expression gene modules defined in DGN dataset. We searched for *trans*-eQTLs among 10,317 SNPs.

The eQTLGen summary statistics are marginal $Z$ scores meta-weighted across multiple cohorts. Most $Z$ scores are from studies where the RNA-seq reads with mappability issues were not filtered out before quantifying gene expression profiles. Therefore, directly applying *trans*-PCO to the summary statistics can lead to false positive signals, which are driven by the cross-mappability between the genes in the module and the *cis*-gene of the test SNP. In order to reduce false positive *trans* signals, we removed from the gene module genes that are cross-mappable to the *cis*-gene (within 100kb) of the test SNP, which is a common practice used in previous studies.[7,16,46] We further removed genes on the same chromosome as the test SNP to prevent the detected *trans* effects from being dominated by *cis* regulations.

The gene expression profiles are not available in eQTLGen. Therefore, to estimate the gene correlation $\Sigma$ of a module, we searched among eQTLGen SNPs for SNPs insignificantly associated with the module (null SNPs) (see Methods S3 for details, Figure S20). We observed that there are less null SNPs that can be found for large modules. And simulations show that the low ratio of the number of

null SNPs used for $\Sigma$ estimation to the module size leads to false positive signals (Methods S3). Therefore, we removed 37 gene modules with ratios lower than 50. Finally, we performed *trans*-PCO on the remaining 129 gene modules.

## QUANTIFICATION AND STATISTICAL ANALYSIS

### Colocalization of *trans*-eQTLs and GWAS loci

To define a region to perform colocalization, we first selected the *trans*-eQTL with the most significant p value and expanded a 200kb flanking genomic region centered at the lead SNP as a region to perform colocalization analysis. We then moved on to the next most significant SNP and expanded a 200kb flanking region. We stopped searching for lead SNPs when all *trans*-eQTLs were included. This resulted in 255 *trans* region-module pairs. As two adjacent regions could correspond to the same colocalization signal, we marked adjacent regions as a region group if their lead SNPs were within 200kb, which generated 179 *trans*-region–module pairs in total. We ran colocalization analysis between each 200kb *trans* region and GWAS loci of 46 complex traits using the R package coloc,[24] assuming there is at most one causal variant for each region. We used the default priors and 0.75 as the PP4 cutoff for significant colocalizations. We defined a merged region group as being colocalized with a trait if any of its 200kb sub-regions has significant colocalization with the trait. We visualized the colocalized regions using LocusCompareR.[47]

### Colocalization of *trans*-eQTLs and *cis*-e/sQTLs

We performed colocalization analysis between *trans*-eQTLs and *cis*-eQTLs (*cis*-sQTLs) of genes near the *trans*-eQTLs. We used the same 179 *trans*-region–module pairs defined in the colocalization analysis of GWAS loci. For a *trans* loci, we searched for the genes within 500 kb around the lead *trans*-eQTLs of the loci, and used these genes to perform colocalization. We used summary statistics of *cis*-eQTLs and *cis*-sQTLs in the DGN dataset from Mu et al.[23] We ran coloc[24] with default priors and 0.75 as PP4 cutoff.

### Trait heritability enrichment in gene modules

To investigate whether a gene module is enriched for trait heritability, we applied stratified LD score regression[42] (S-LDSC) to 166 co-expression gene modules and 46 complex traits and diseases. Specifically, for each module we defined the annotation set as the SNPs within genomic regions of genes in the module and also a 500 base-pair window around the genes. We also included 97 annotations from the baseline model. Partitioned heritability enrichment was calculated as the proportion of trait heritability contributed by SNPs in the module annotation over the proportion of SNPs in that annotation.

### Association of drug targets with disease-relevant gene sets regulated in *trans*

To show the translational application of *trans*-PCO results, we examined whether drug targets are more likely to be associated with disease-relevant pathways or gene sets in *trans* (Methods S4). We first downloaded drug targets of various diseases from The Broad Institute Drug Repurposing Hub (https://repo-hub.broadinstitute.org/repurposing). We then examined whether the drug targets are near any SNPs that have significant trans associations with immune-related gene co-expression modules or hallmark gene sets in the eQTLGen dataset.

