## [Document S2. Transparent peer review records for Wang et al · Cell Genomics]

Trans-eQTL mapping in gene sets identifies network effects of genetic variants

Lili Wang, Nikita Babushkin, Zhonghua Liu, Xuanyao Liu

Summary

Initial submission: Received : May 10, 2023

Scientific editor: Sara Rohban

First round of review: Number of reviewers: 3
Revision invited : Dec 06, 2023
Revision received : Dec 08, 2023

Second round of review: Number of reviewers: 2
Revision invited : Jan 16, 2024
Revision received : Feb 07, 2024

Third round of review: Number of reviewers: 1
Accepted : Mar 13, 2024

Data freely available: YES

Code freely available: YES

This transparent peer review record is not systematically proofread, type-set, or edited. Special characters, formatting, and equations may fail to render properly. Standard procedural text within the editor's letters has been deleted for the sake of brevity, but all official correspondence specific to the manuscript has been preserved.

Referees' reports, first round of review

Reviewer #1:

The authors describe a method to group sets of co-expressed and test whether individual variants affect these groups of co-expressed genes. The authors subsequently apply the methodology to the DGN dataset and subsequently to summary statistics from eQTLGen. The authors show several examples of mostly blood cell-type component associated variants that affect these groups of co-expressed genes.

I do appreciate that the authors use this line of reasoning, but I am somewhat surprised that the authors do not compare their method to several methods that have been presented before. I discuss some of these below:

- Kolberg et al, eLife 2020 state in their abstract: "We used co-expression modules inferred from gene expression data with five methods as traits in trans-eQTL analysis to limit multiple testing and improve interpretability".
- Dutta et al, Nature Communications 2022, also use DNG and eQTLGen and state in their abstract: "identify sets of gene-expressions trans-regulated by sets of known trait-related genetic variants"
- Rotival et al, PLOS Genetics 2011 (12 years ago) state in their abstract: "We applied a method of extraction of expression patterns-independent component analysis to identify sets of co-regulated genes. These patterns were then related to 675,350 SNPs to identify major trans-acting regulators".

The authors do refer to some of these papers and also several other methods, but do not compare their method against these earlier methodologies. I find this particularly troublesome for the Dutta et al, Nature Communications 2022 paper, since the authors of that paper used exactly the same datasets, and thus it should be possible to establish a fair comparison. That being said: It might be that I misinterpret this paper, and do not understand why the goal of this paper is completely different as compared to the aforementioned papers. However, while the authors in the introduction of the manuscript state that their approach supposedly has some benefits to other methods, no comparison is shown, nor is there any reflection on this in the discussion section.

As such I regretfully am unable to determine what is the novelty of this approach, and I consider this essential for the field to get guidance on how to do these kinds of analyses in the future with ever larger molecular QTL studies emerging in the field.

Reviewer #2:

This is a well done paper that proposes a method for testing for trans-associations for SNPs with gene-expression networks through underlying principal components - the advantage being reducing the number of tests and aggregating signals across genes. The methods and data analyses pipelines used are generally solid and the paper is well written. Here are my comments

1) The authors have proposed combing an array of six different PC-based tests to maximize the power of detecting genetic associations. I could not find easily what method they are proposing to calculate the p-value for such an omnibus test.

2) Dutta et al (Nature Communication, 2021) had proposed another aggregative method for detecting association between networks of SNPs and gene-expressions. The method was mentioned cursorily in the introduction, but not otherwise. Given the goal of these two papers are overlapping, I was hoping the authors would have made more direct contrast of these two methods, including in simulation studies and data analyses.

3) One major limitation of the approach is, as acknowledged in the discussion, is that it does not detect individual genes in a network which is driving the trans-association signal. I feel the user, however, can use some exploratory tool to prioritize genes in a networks that are key drivers for the underlying signal.

4) For trans-loci that colocalize with complex traits, it would be of further interest to explore evidence of causality of the identified gene expression network on the complex trait using the MR framework. Here cis-eQTLs for the genes in the various modules could be used as the underlying instruments for testing evidence of their causal effects on the complex trait. As there are many genes in a module, there needs to be a way of first generating a set of top gene list which is driving the trans-signals (see my comment 3). Then one could test whether cis-eQTL for these top genes show any evidence of association with the underlying complex trait. Being able to show identified gene-networks have causal effects on complex traits will have stronger downstream impact of this work such as for drug development or repurposing, which is one of the ultimate translational application of all these types of work.

5) I would like to see a stronger discussion at the end about the translational applications of the proposed work on complex trait genetics (e.g drug development, risk prediction etc).

Reviewer #3:

This paper proposed trans-PCO, a principal component-based multivariate association test for trans-eQTL. From the simulation studies and real data analysis, the authors showed that the multivariate approach can be more powerful than the univariate approach. Multivariate tests in genetics and genomics have been studied extensively, and it seemed like this a nice application of the multivariate test. I have some comments.

Comment

1. The multi-variate test is a nice approach to improve power; however, it is not always more powerful than a single variate test. If the assumption is not true (ex., High level of sparsity, low-level of correlation among variates), it can have a lower power than a single variate test. In the current simulation, since the effect sizes are small, all methods have very low power when the sparsity level is high (low causal proportion). The authors need to consider a situation of high-level sparsity and high effect size (like change sparsity level in Supplementary Figure S1)

Or also can consider a larger sample size (if you think increasing effect size in high sparsity setting is artificial), as some studies (like eQTLGen) have quite a large sample size.

2. As the authors discussed, after the multivariate test, one of the important issues is to identify the gene-snp pairs that drive the signal. I am wondering whether the single variate test can help to figure this out.

3. Violation of the assumptions in the multivariate test can cause type I error inflation (for example, multivariate normal distribution (MVN) assumption). If the data does not follow MVN, can the method control type I error rates?

4. One of the advantages of the proposed approach the authors mentioned in the paper is the careful removal of misaligned reads.

“we filtered out the reads that were mapped to multiple genomic regions and reads with >2 mismatches”.

But in summary statistics-based analysis (eQTLGen),

“we removed the genes in a module that are cross-mappable with any cis-genes (genes within 100 kb) of the test SNP”.

It doesn't look like they are consistent, as the second seems more like removing cis-effects. The authors need to provide a more detailed explanation of why the second (summary-based) QC is good enough.

5. Tpc0 is the minimum p-values of several PC-based test approaches, which is not the p-value. Description of p-value calculation of Tpc0 will be helpful for readers.

Minor comment

1. Figure 2: I cannot see error bars. Is it because of error bars are too small?

Authors' response to the first round of review

We want to thank the reviewers for their helpful and valuable comments, which have significantly improved our manuscript in terms of completeness and clarity. Below are our point-to-point responses (colored in blue) to these comments. We also highlight revisions in blue made in the manuscript.

Reviewer #1

The authors describe a method to group sets of co-expressed genes and test whether individual variants affect these groups of co-expressed genes. The authors subsequently apply the methodology to the DGN dataset and subsequently to summary statistics from eQTLGen. The authors show several examples of mostly blood cell-type component associated variants that affect these groups of co-expressed genes.

I do appreciate that the authors use this line of reasoning, but I am somewhat surprised that the authors do not compare their method to several methods that have been presented before. I discuss some of these below:

- Kolberg et al, eLife 2020 state in their abstract: "We used co-expression modules inferred from gene expression data with five methods as traits in trans-eQTL analysis to limit multiple testing and improve interpretability."
- Dutta et al, Nature Communications 2022, also use DNG and eQTLGen and state in their abstract: "identify sets of gene-expressions trans-regulated by sets of known trait-related genetic variants"
- Rotival et al, PLOS Genetics 2011 (12 years ago) state in their abstract: "We applied a method of extraction of expression patterns—independent component analysis—to identify sets of co-regulated genes. These patterns were then related to 675,350 SNPs to identify major trans-acting regulators."

The authors do refer to some of these papers and also several other methods, but do not compare their method against these earlier methodologies. I find this particularly troublesome for the Dutta et al, Nature Communications 2022 paper, since the authors of that paper used exactly the same datasets, and thus it should be possible to establish a fair comparison.

That being said: It might be that I misinterpret this paper, and do not understand why the goal of this paper is completely different as compared to the aforementioned papers. However, while the authors in the introduction of the manuscript state that their approach supposedly has some benefits to other methods, no comparison is shown, nor is there any reflection on this in the discussion section.

As such I regretfully am unable to determine what is the novelty of this approach, and I consider this essential for the field to get guidance on how to do these kinds of analyses in the future with ever larger molecular QTL studies emerging in the field.

Response We sincerely thank the reviewer for these comments and we completely agree that comparison to these methods is essential and necessary for evaluating our work. Therefore, in our revised manuscript, we provided thorough comparisons to the three methods in simulations and/or real data analyses. We provide the details of the comparisons and results for each method in this response letter as well as the revised manuscript.

1. Kolberg et al., eLife 2020.

We want to point out that the comparison to Kolberg et al.¹ in both simulations and real data was included in our original manuscript, but we labeled the method as “PC1-method”, rather than directly referring to it as the Kolberg et al. method, which was confusing. The Kolberg et al. method applies the similar idea of first constructing co-expression modules and then testing associations between a variant and a group of genes. The method essentially uses the first primary PC (PC1) of a gene module to represent the co-expression pattern among genes. Specifically, the method first infers gene co-expression modules using two types of methods - co-expression clustering (WGCNA and funcExplorer) or matrix factorisation (PLIER, ICA and PEER) methods. Each gene co-expression module is represented by a single ‘eigengene’ that captures the co-expression correlation within the module. As noted in Kolberg et al.¹, the ‘eigengene’ is essentially PC1 of a gene module or highly correlated with PC1.

However, previous studies have shown that PC1 has very limited power in identifying genetic effects on multiple genes^{2,3}, even though PC1 captures the largest amount of total variance in gene expression levels. Higher order PCs may have better power than PC1,

but it is hard to predict which PCs to use to achieve the best power, as achieving high powers depends on not only a good representative of a gene module, but also the true genetic effects of variants on genes in the module, which is unknown in practice³. Therefore, to achieve high and robust power, we used an omnibus PC-based test (PCO), which uses multiple PCs and combines these PCs in several linear and nonlinear ways to capture various genetic effects under different genetic architectures³.

We performed a thorough comparison of our method and Kolberg et al.¹ both in simulation (Figure 2, Figure S1, Figure S14, Figure S18, Methods 'Simulation') and real data analysis (Figure S13, Results in main text Page 8, Lines 284-301, Supplementary Note 'Apply the primary PC method to DGN dataset') in the original version of the manuscript. In simulations, we showed that the PC1 approach (i.e. Kolberg et al) indeed had limited power for detecting *trans*-eQTLs associated with multiple genes' expression levels. For example, in Figure 2, the PC1 approach had a power of 0.0018% versus 74% of trans-PCO at the sample size of 800 under a specific but realistic simulation setting. In Figure S1, we showed that PC1 also had very limited power (near 0% power) in comparison to trans-PCO and the univariate approach under various simulation settings of genetic variance. We also applied the Kolberg et al. method to the real dataset DGN. In summary, Kolberg et al. method identified a much less number of *trans* signals than trans-PCO. More specifically, it identified 1483 significant *trans*-eSNP–module pairs (55 *trans*-loci–module pairs) at 10% FDR, and 1464 pairs (99%) were detected by trans-PCO (Figure S13). Overall, Kolberg et al. method identified 38% of *trans* signals detected by trans-PCO (Figure S13).

While this result supports that trans-PCO is more powerful than Kolberg et al. method, there were more signals detected by Kolberg et al. method in real data than expected from simulation results, where it remains powerless across almost all simulation settings. To address the discrepancy between the Kolberg et al. method performance in simulations versus real data, we performed additional simulations and analyses with real data (Figure S13, Figure S14, Figure S18).

We note that we simulated weak *trans* effects and sparse causal proportions in the simulations in order to better reflect common and realistic *trans* effects. To further represent general use cases, we did not specify the direction of the genetic effects on genes in a module in relation to the primary PC of the module. Although Kolberg et al. method remains powerless in these general simulation settings, we showed in additional simulations that it can have good statistical power in some specific cases. For example, we performed additional simulations, where the effects of a genetic variant on genes in a module align with the primary PC direction of the gene module, and we found that Kolberg et al. method has very high power (Figure S14). However, it is impossible to predict whether the directions would align in real data as the true genetic effects are unknown.

Additionally, we found that the Kolberg et al. method also gains power when the genetic effects are substantially large and the proportion of causal genes in the gene module is high. We performed simulations, where the genetic variance is 0.2 and the proportion of causal genes is 100%, the method has 50% power of identifying signals (Figure S18).

Therefore, we expect some *trans* signals that are detected by the Kolberg et al. method in real data are likely to be among the strongest *trans* effects. We compared the univariate z-scores of *trans*-eQTLs detected by trans-PCO and the Kolberg et al. method (Figure S13B-D). We found that signals identified by the Kolberg et al. method have higher z-scores than signals detected by trans-PCO, supporting that the Kolberg et al. method detected those strong *trans* effects in real data, whereas trans-PCO detected additional *trans* signals with much weaker effects.

We did not apply the Kolberg et al. method to eQTLGen dataset, because their method is not applicable to summary-statistics level data. In contrast, trans-PCO can be applied to both raw individual level expression/genotype dataset and summary-statistical level data.

Through simulations and real data analyses, we conclude that trans-PCO is much more powerful than Kolberg et al. at identifying *trans*-eQTLs of gene sets. In addition, trans-PCO has wider applications as it does not require individual level data of gene expression and genotypes.

We originally included these results and discussion in Supplementary Note and Figures. To reflect the revisions and make the method comparison more clear, we added more details and moved the discussion from Supplementary Note to the main text (Page 8, Lines 284-301), which now reads,

“We also applied the primary PC method (Kolberg et al.²³) to DGN, and identified 1483 significant *trans*-eSNP–module pairs (55 *trans*-loci–module pairs) at 10% FDR, and 1464 pairs (99%) were detected by trans-PCO (Figure S13A). Notably, in total, trans-PCO identified more than twice the signals than the primary PC method. However, the primary PC method identified more signals than expected, as it was previously shown to be powerless in the simulations. We note that we simulated weak effects and sparse causal proportions to better reflect common and realistic *trans* effects, and the primary PC method is powerless in these settings. We performed additional simulations with large effects and high causal proportions, and the primary PC method achieved 50% power as trans-PCO (see Supplementary Note, Figure S18). Additionally, we found in the DGN dataset that the univariate z-scores of *trans* signals detected by the primary PC method are larger than those of trans-PCO signals (Figure S13B-D). Therefore, the *trans* signals detected by the primary PC method are likely of strong *trans* effects, and

trans-PCO is able to detect additional weak *trans* effects. Statistically, PC1 can also have good statistical power when the *trans* effects align with the direction of PC1²⁶. To demonstrate this, we performed simulations under the assumption that the genetic effect vector perfectly aligns with the primary PC direction. We found that the primary PC method has higher power than trans-PCO and MinP methods under this special circumstance (Figure S14). However, it is impossible to predict when the genetic effects align with gene expression PCs.“

To further make it clear that the PC1 (or primary) method refers to the Kolberg et al. method, we added more details in introduction (Page 3, Lines 104-107), and annotated the PC1 method as Kolberg et al. throughout the manuscript and supplementary material, for example, Page 6, Line 200.

2. Dutta et al., Nature Communications 2022

We sincerely thank the reviewer for suggesting the comparison to this method. We agree that ARCHIE proposed in Dutta et al.⁴ and trans-PCO are two seemingly similar methods. However, we note that these two methods have different goals and usages and they are not directly comparable (see below). Yet, to give readers and users insights on when and how each method should be used, we thoroughly compared ARCHIE and trans-PCO in both simulations and real data analyses using three strategies. We elaborate our points as below.

ARCHIE stands for Aggregative tRans assoCiation to detect pHenotype specific gEne-sets, which identifies sets of distal genes whose expression levels are *trans*-regulated by sets of known trait-related genetic variants, using sparse canonical correlation analysis. ARCHIE and trans-PCO are designed with different goals and usages. We emphasize four main differences between two methods. First, ARCHIE captures *trans* regulations that reflect only trait-specific associations. Specifically, ARCHIE tests significance of an observed canonical correlation (cc-value), which quantifies association between a gene set and a variant set, against a competitive null hypothesis, which uses cc-values of all eQTLGen trait-associated variants as empirical null distribution and reflects *trans* regulations not specific to any trait⁴. As a result, an ARCHIE p-value reflects significance of trait-specific patterns. In contrast, trans-PCO identifies *trans*-eQTLs under the general null hypothesis assuming no *trans* effects. Therefore, trans-PCO can be used to generate comprehensive maps of *trans*-eQTLs in tissues and cell types, in non-trait-specific manner. Using ARCHIE to perform genome-wide scan of *trans*-eQTLs in a non-trait specific manner can be challenging, as non-trait specific p-value is not computable in current implementation of the method and it will be extremely computational challenging (due to the computational intensive resampling procedure and difficulty of manipulating

whole-genome LD matrices⁴). Second, trans-PCO and ARCHIE are designed to capture different *trans* regulatory effects. ARCHIE is powerful when multiple trait-associated variants have weak effects on a single gene (for example, multiple GWAS loci converge on the core genes through *trans* regulation; Figure 1A below and Figure 2 of the ARCHIE paper by Dutta et al.⁴). In contrast, trans-PCO is designed to capture weak *trans* signals of a variant on multiple co-regulated genes, for example, a transcription factor has *trans* effects on multiple target genes (Figure 1B below). Third, *trans* signals captured by ARCHIE are hard to be used to study *trans* regulatory mechanisms. ARCHIE identifies components, consisting of multiple trait-associated SNPs and multiple genes, where sets of gene expressions are *trans* regulated by sets of trait-associated variants. Without knowing the exact *trans*-eQTL SNP driving the *trans* regulation, it is hard to explore the mechanism of the *trans*-eQTL, for example, whether the *trans*-eQTL is also a *cis*-eQTL, or which gene is the *trans* regulator. Fourth, ARCHIE takes all genes as input and infers gene sets as gene components, whereas trans-PCO is flexible to be applied to any user-defined gene set of interest.

A. ARCHIE identifies weak *trans*-eQTL effects from multiple trait-associated SNPs to genes

B. trans-PCO identifies weak *trans*-eQTL effects from a single variant to multiple co-regulated genes

Figure 1. ARCHIE and trans-PCO are designed to detect different *trans*-eQTL signals. (A) ARCHIE detects weak *trans*-eQTL effects from multiple trait-associated SNPs to genes, for example, multiple GWAS SNP to a “core” gene or a “core” process. (B) trans-PCO identifies weak *trans*-eQTL effects from a single SNP to multiple co-regulated genes, for example, from a master regulator to multiple regulated genes.

The differences in goals and usages of ARCHIE and trans-PCO make them not directly comparable. Yet, we provided thorough comparisons of ARCHIE and trans-PCO using both simulations and real data analyses with the following three strategies -

(1) In simulations, we evaluated how well ARCHIE can detect regular *trans*-eQTL signals between a single SNP and a gene set. We used a real co-expression gene module consisting of 101 genes from DGN dataset to simulate data. We simulated z-scores from a normal distribution using the correlation matrix of the gene module, sample size 500, causal gene proportion 30% with 30 genes being true target genes, and genetic variance

0.001 as parameters. More detailed simulation procedure was described in Methods “Simulation”. Trans-PCO has a power of 36% under this setting.

To run ARCHIE, three main inputs are needed⁴, including (1) Σ_{GG} , column-correlations of genetic variants, (2) Σ_{EE} , column-correlations of gene expressions, and (3) Σ_{GE} , cross-covariance matrix between variants and gene expressions. In our simulation settings, we tested variant at a time. Therefore, $\Sigma_{GG} = 1$. We set $\Sigma_{EE} = \Sigma_{101}$, and we approximate Σ_{GE} with Z_{101}/\sqrt{N} . We used various genetic variances ranging from 0.002 to 0.006. For each scenario, 1000 simulations were performed.

ARCHIE calculated canonical correlations (cc-values, Figure S19A, see below) that measure the *trans* association between each variant and gene sets across 1000 simulations and scenarios. ARCHIE also selected genes (Figure S19B) from the 101 input genes to be target genes in ARCHIE components. While only 30% of genes have true *trans* effects, ARCHIE selected nearly all 101 genes in the ARCHIE component (Figure S19B). To calculate the p-value of a cc-value, we simulated an empirical null distribution of cc-values by simulating one million null z-scores. Then, an empirical p-value is calculated to be the expected number of null cc-values larger than the observed cc-value (Figure S19C). We found that ARCHIE p-values are deflated across genetic variances, and it was not able to identify significant tests in any of 1000 simulations in our simulation settings, indicating ARCHIE’s lower power to detect *trans*-eQTL signals from a single SNP to multiple co-regulated genes (Figure S19D).

Figure S19. Comparison of trans-PCO and ARCHIE in Dutta et al.⁸. (A)-(D) Simulation comparison. (A) Distribution of ARCHIE cc-values across various genetic variances. (B) Number of selected genes by ARCHIE across various genetic variances. Red line indicates the number of true target genes, i.e. 30. (C) QQ plot of empirical p-

values across various genetic variances. (D) Distribution of adjusted p-values across various genetic variances. Red line shows FDR level 0.05.

In the Dutta et al. manuscript, they performed simulations to evaluate the performance of ARCHIE, following models similar to Figure 1A (above). However, the simulation was complicated and no source code was released. Therefore, we did not apply trans-PCO to their simulations, instead we came up with two strategies to evaluate how well trans-PCO could replicate *trans*-eQTL signals identified by ARCHIE in real data analyses of eQTLGen.

(2) In eQTLGen, we evaluated how well trans-PCO can replicate signals detected by ARCHIE. ARCHIE identified gene sets that are significantly associated with disease-associated variants of 29 traits using eQTLGen summary statistics, though only results for three traits were publicly available. There were five significant ARCHIE components identified for the three traits. Specifically, 2 (resp. 1 and 2) components were identified to have significant *trans* associations with prostate cancer (resp. schizophrenia and ulcerative colitis)-associated variants. Each component contains a set of SNPs and a set of genes, and the set of SNPs are correlated to the set of genes through *trans* regulation. We applied trans-PCO to eQTLGen summary statistics and used the five ARCHIE-selected gene sets as gene modules. We performed trans-PCO on the five gene sets and calculated their association p-values with each single eQTLGen variant. P-values were adjusted for multiple testing to control the false positive rate by Bonferroni correction (FDR < 0.05). The goal is to evaluate whether trans-PCO can replicate the *trans*-eQTL signals identified by ARCHIE.

Among the five components, four had significant *trans*-associations by trans-PCO with at least one of ARCHIE selected variants in the same component (Figure S19G, below). Therefore, trans-PCO replicated 80% of the *trans*-signals identified by ARCHIE at 5% FDR. In total, ARCHIE identified 134 variants in the five components. Trans-PCO replicated a large proportion (min:36.5%~max:85.1%) of the variants for component 1's (C1's) for the three diseases (Figure S19G). Only one out of the thirteen variants (7.7%) were replicated by trans-PCO for component 2 (C2) of prostate cancer. These results indicate that trans-PCO is well powered to replicate *trans* signals in component 1's of ARCHIE components, but not component 2's. Nonetheless, trans-PCO identified 1655 additional significant *trans*-eQTL SNPs at 5% FDR, ranging from 65 to 640 for each gene set (Figure S19F).

Figure S19. Comparison of trans-PCO and ARCHIE in Dutta et al.⁸. (E)-(G) Trans-PCO on ARCHIE selected gene sets. (E) X-axis shows significant ARCHIE components for three traits, prostate cancer (Prosc), Schizophrenia, and Ulcerative Colitis (UC). C1 and C2 mean the first and second component. Each component is a pair of selected genes set and variants set. Y-axis shows the size of ARCHIE gene sets (light) and the number of independent null variants (dark) used to estimate correlation matrix of the gene sets. (F) P-values (with Bonferroni correction) of each variant and ARCHIE gene set by trans-PCO. Each panel is an ARCHIE component. Red cross indicates variants selected by ARCHIE. Grey line shows FDR level 0.05. (G) Number of ARCHIE variants across components that are also significant by trans-PCO (dark).

(3) We directly compared *trans* signals reported by trans-PCO and ARCHIE in the eQTLGen datasets. In summary, ARCHIE identified gene sets that are significantly associated with disease-associated variants of 29 traits, though only results for three traits were publicly available. There were five significant ARCHIE components for the three traits. We also applied trans-PCO to eQTLGen summary statistics as described in the section “Summary-statistics-based trans-PCO identified 10,167 *trans*-eSNP-module pairs in eQTLGen” in the original manuscript. We analyzed 129 co-expression gene modules and identified 8116 significant *trans*-eSNP-gene co-expression module pairs, corresponding to 2161 eQTLGen test SNPs and 122 gene modules.

To check if ARCHIE signal components found in eQTLGen were replicated by trans-PCO, we checked (1) if ARCHIE selected genes are included in the significant *trans* target gene modules identified by trans-PCO (Figure S19H, below), (2) if ARCHIE selected variants are replicated as *trans*-eQTL SNPs by trans-PCO (Figure S19I). Among selected genes by ARCHIE, all of those included in our eQTLGen analysis were included in a significant

trans-eQTL module by trans-PCO (Figure S19H). Among selected variants by ARCHIE, 31% (40 out of 129 included variants) were also replicated as significant *trans* signals by trans-PCO (Figure S19I). Additionally, trans-PCO identified 15x more significant *trans*-eQTL SNPs than ARCHIE. These observations support that while both ARCHIE and trans-PCO can identify weak *trans*-effects, they are designed to detect different *trans* signals; ARCHIE is designed to identify the *trans* signals in which a target gene has weak *trans*-effects from multiple SNPs (for example from multiple GWAS SNPs to a single core gene) as modeled in Dutta et al., but trans-PCO is designed to detect *trans* effects from one SNP to multiple genes (for example the *trans* effects from a master regulator to multiple downstream genes).

Figure S19. Comparison of trans-PCO and ARCHIE in Dutta et al.⁸. (H)-(I) Comparison of eQTLGen signals by trans-PCO and ARCHIE. (H) Number of selected genes across ARCHIE components. Lightest blue (Selected) represents the number of selected genes by ARCHIE of each component. Darker blue (Included) represents the selected genes included in trans-PCO eQTLGen analysis. Darkest (Signal) represents genes included in a significant *trans* target gene module by trans-PCO. All ARCHIE selected genes that are analyzed by trans-PCO are in significant *trans* gene modules by trans-PCO. (I) Number of selected variants across ARCHIE components. Labels are similar as in (H).

Lastly, we note that Dutta et al.⁴ did not analyze DGN dataset to identify sets of genes *trans* regulated by sets of trait-relevant variants. Instead, DGN dataset was only used as an alternative reference dataset to estimate the co-expression matrix, in order to evaluate the robustness of ARCHIE results to the estimation of co-expression matrix. Specifically, the implementation of ARCHIE requires three input matrices, one of which is co-expression matrix, which quantifies the correlation among gene expression levels. To estimate the co-expression matrix, the way used throughout their paper was to use gene expression levels of individuals in GTEx v8 data of whole blood. To investigate the robustness of ARCHIE results to co-expression estimation, they used gene expression

levels from another reference dataset DGN. We therefore were not able to compare trans-eQTL results in DGN dataset.

In summary, through simulations and real data analysis, we demonstrated that trans-PCO and ARCHIE are designed to detect different types of weak *trans*-eQTL signals. The main goal of ARCHIE is to identify trait-specific gene sets associated with GWAS loci, whereas trans-PCO is designed to map *trans*-eQTLs for any user-specified gene sets in specific tissues or cell types. Therefore, if the goal is to comprehensively map *trans*-eQTLs for any gene sets of interest, trans-PCO should be the method of choice, as it identifies a lot more *trans* signals as shown in both simulations and eQTLGen analyses. Additionally, trans-PCO could also be used to identify disease relevant genes and processes through follow up analyses, such as colocalization analyses.

To reflect the revisions, we added more content about ARCHIE in the Introduction (Page 3, Lines 108-111), Results (Page 7, Line 251-259), and Discussion in the main text (Page 18, Lines 593-636). We also included all the detailed comparisons performed in simulations and read data in Supplementary Note (“Compare trans-PCO and ARCHIE”, Page 5-10, Lines 149-329). We also added Figure S19 as a Supplementary figure.

3. Rotival et al., PLOS Genetics 2012

Rotival et al.⁵ proposed a method to identify *trans*-eQTLs of co-expressed gene sets, which shares the same goal as trans-PCO. However, we used simulations to demonstrate the Rotival et al. method has minimal power to identify weak *trans* effects. We will first describe how the Rotival et al. method works and then demonstrate the performance of the method in simulations.

The Rotival et al. method consists of a few main steps: (1) Independent Component Analyses (ICA), an matrix factorisation method, is used to infer components representing co-expression patterns from the expression of all genes; (2) ICA components were then tested against all SNPs to filter out non-suggestive associations (p value $> 1e-7$); (3) for the remaining components and SNPs, a subset of genes contributing strongly to each component were selected as a gene module; (4) for a pair of gene module and a SNP, a significant association is identified if the genes in the module are enriched in genes that are individually associated to the SNP (univariate p -value $< 1e-5$) compared to all other background genes outside the module. The enrichment was tested using a hypergeometric test. We want to point out that the Rotival et al method does not have any available software or code released. We performed the comparisons based on the description of the methods in their paper.

We want to note two points in the comparison of Rotival et al. method and trans-PCO. First, filtering out non-suggestive associations in step (2) can lead to loss of power for identifying *trans*-eQTLs. Second, enrichment analysis to quantify the associations between a gene module and a SNP has limited power at detecting weak *trans* signals. We elaborate our points as follows.

First, we argue that filtering associations of ICA components and SNPs in step 2 can be a major power limiting step. To calculate the association between an ICA component and a SNP, the factor loadings of the ICA component are used as the component (or module) profile. As observed in Kolberg et al.¹, the factor loadings of matrix factorisation (or eigengenes) is highly correlated with the first primary PC (PC1) of gene modules defined by co-expression clustering analysis. However, we and others have shown that PC1 has very limited power for detecting *trans* genetic effects between co-regulated gene sets and variants (see more discussions on PC1 having limited power in our response to comparison with Kolberg et al.). Therefore, many *trans* signals would have weak associations with PC1, and thus would be removed from the remaining signals used in following steps to identify final *trans* signals.

Additionally, we note that the enrichment analysis by hypergeometric test is not powerful at detecting weak *trans* effects. As stated above, Rotival et al. essentially uses hypergeometric tests to identify *trans*-eQTL SNP-component associations, which are expected to have an enrichment of weak *trans* effects. We therefore performed simulations to evaluate the performance of the enrichment test used by Rotival et al., assuming genes representing co-expression module are already known. The simulations were adapted from the original simulations evaluating the power of trans-PCO as described in Methods (“Simulation”). We first simulated the z-scores between a SNP and $K = 101$ genes in a gene module, following the distribution $N_K(\sqrt{n}\beta, \Sigma_{K \times K})$, where n is sample ($N = 500$), β is a vector representing the true effect sizes of the SNP on K genes and $\Sigma_{K \times K}$ is the residualized expression correlation matrix of 101 genes from a real gene module of DGN dataset. Among K genes a proportion γ of them are causal with non-zero effects. Therefore, we generated β_k from a point normal distribution, where $\beta_k \sim N(0, \sigma^2_b)$ for proportion γ ($\gamma = 1\%, 5\%, 10\%, 30\%$ and 50%), and $\beta_k = 0$, otherwise. The *trans* genetic variance σ^2_b is set to be 0.001 as default. We also tried larger variances, 0.01, 0.05, 0.1, and 0.2. We simulated 10k SNPs for each simulation and 1k simulations.

To check if target causal genes are enriched in genes included in the module, we also simulated the “background” genes, i.e. genes outside the module. We assume genes outside the module are independent and there are no target causal genes. We simulated 12,001 background genes (12,102 genes used in DGN dataset, subtracted by 101 genes included in the module) from standard normal distribution with zero effects. To define significant individual associations for enrichment, we used p-value cutoff $1e-5$ to be

consistent with Rotival et al.. Then the enrichment p-value of a SNP for the gene module was calculated by the hypergeometric test. P-values were adjusted using 'qvalue' at $FDR < 0.1$. Power was calculated as the proportion of SNPs that were identified to be significant among 10k SNPs.

As shown in Figure S20 below, Rotival et al. has minimal power at detecting *trans* associations in the case of weak effects. Under the setting where genetic variance is 0.001, the enrichment test has no power across all causal proportions, while trans-PCO has much higher power, for example, power is 37% when 30% genes are true target genes. We note that the enrichment test can have power for detecting *trans*-eQTLs when the *trans* effects are large (which is not common for *trans*-eQTLs). For example, when genetic variance is 0.01, the enrichment test has a power of 32% when 5% genes are target causal genes. Trans-PCO has a higher power of 64% under the same setting. In the case of even larger genetic variances, e.g. 0.1 and 0.2, the enrichment test has a comparable power with trans-PCO. In summary, the enrichment test does not have power to detect multiple weak *trans* effects.

To reflect the revisions, we added more details in the introduction (Page 3, Lines 101-103), Results (Page 7, Line 257-259), and a section in Discussion (Page 19, Lines 636-640), and a section in Supplementary Note ("Compare trans-PCO and Rotival et al.", Page 9, Lines 330-398). We also added Figure S20 as a Supplementary figure.

Figure S20. Comparison of trans-PCO and Rotival et al.¹⁰. We compared the power of trans-PCO and the method proposed in Rotival et al. across various causal proportions under different genetic variances. Specifically, we simulated the proportion of target genes with non-zero effects to be (1) high levels of sparsity 1%, 5%, 10%, and (2) low levels of sparsity 30%, and 50%, and genetic variances to be (1) small effect 0.001, and (2) large effect 0.01, 0.05, 0.1, and 0.2. We used the same gene module as in Figure 2 and simulated the sample size to be 500. Rotival et al. method used a hypergeometric test to calculate enrichment p-values. P-values were corrected by 'qvalue' to control false positive rate at 10%. Power was computed from 10k SNPs across 1000 simulations.

In summary, we sincerely thank the reviewer again for the comments on method comparisons, which have significantly improved our study. By thoroughly comparing the performance of trans-PCO versus the three methods, it is clear that trans-PCO significantly outperforms the existing approaches for mapping *trans*-eQTLs of coexpressed gene sets. While the co-regulation patterns of genes by *trans*-eQTLs have long been recognized, trans-PCO provides a powerful and elegant solution to mapping *trans*-eQTLs of these gene sets. Trans-PCO method was carefully made into an easy-to-use and reproducible pipeline. It is flexible and can be applied to both RNA-seq data with genotypes or summary statistics, and the user can define various gene sets (e.g. biological pathways or processes) as modules of interest. Our map of *trans* effects can also be used in follow up analysis, such as colocalization, to improve our understanding how trait associated loci impact gene regulatory networks and pathways through *trans* regulatory effects.

Reviewer #2

This is a well done paper that proposes a method for testing for *trans*-associations for SNPs with gene-expression networks through underlying principal components - the advantage being reducing the number of tests and aggregating signals across genes. The methods and data analyses pipelines used are generally solid and the paper is well written. Here are my comments

1) The authors have proposed combining an array of six different PC-based tests to maximize the power of detecting genetic associations. I could not find easily what method they are proposing to calculate the p-value for such an omnibus test.

Response We thank the reviewer for the comment. To calculate the p-value of the final omnibus test statistics,

$$T_{PCO} = \min p_{\{PCMinP, PCFisher, PCLC, WI, Wald, VC\}}$$

we first take inverse-normal transformation of the test statistics to transform it to be normally distributed and calculate the p-value as,

$$p_{TPCO} = 1 - P\{\min \Phi^{-1}(p_{\{PCMinP, PCFisher, PCLC, WI, Wald, VC\}}) > \Phi^{-1}(T_{PCO}^{obs})\}$$

where Φ^{-1} denotes the inverse standard normal cumulative distribution function. Then the p-value can be efficiently computed using a multivariate normal distribution of six different PC-based tests as described in Liu et al.³.

To make the p-value computation clear in the manuscript, we added more details in the Method section (Page 22, Lines 749-754). The text now reads:

“The p-value of PCO test statistics can be computed by performing an inverse-normal transformation of the test statistics,

$$p_{TPCO} = 1 - P\{\min \Phi^{-1}(p_{\{PCMinP, PCFisher, PCLC, WI, Wald, VC\}}) > \Phi^{-1}(T_{PCO}^{obs})\}$$

where Φ^{-1} denotes the inverse standard normal cumulative distribution function. The p-value can be efficiently computed using a multivariate normal distribution as described in Liu et al.²⁵.”

2) Dutta et al (Nature Communication, 2021) had proposed another aggregative method for detecting association between networks of SNPs and gene-expressions. The method was mentioned cursorily in the introduction, but not otherwise. Given the goal of these two papers are overlapping, I was hoping the authors would have made more direct contrast of these two methods, including in simulation studies and data analyses.

Response We sincerely thank the reviewer for the suggestion. We agree that ARCHIE proposed in Dutta et al.⁴ and trans-PCO are two seemingly similar methods. Therefore, we carefully and thoroughly compared the two methods in both simulations and in real data analyses in the revision. However, we first would like to note that these two methods have different goals and usages and they are not directly comparable.

ARCHIE stands for Aggregative tRans assoCiation to detect pHenotype specific gEne-sets, which identifies sets of distal genes whose expression levels are *trans*-regulated by sets of known trait-related genetic variants, using sparse canonical correlation analysis. ARCHIE and trans-PCO are designed with different goals and usages. We emphasize four main differences between two methods. First, ARCHIE captures *trans* regulations that reflect only trait-specific associations. Specifically, ARCHIE tests significance of an observed canonical correlation (cc-value), which quantifies association between a gene set and a variant set, against a competitive null hypothesis, which uses cc-values of all eQTLGen trait-associated variants as empirical null distribution and reflects *trans* regulations not specific to any trait⁴. As a result, an ARCHIE p-value reflects significance of trait-specific patterns. In contrast, trans-PCO identifies *trans*-eQTLs under the general null hypothesis assuming no *trans* effects. Therefore, trans-PCO can be used to generate comprehensive maps of *trans*-eQTLs in tissues and cell types, in non-trait-specific manner. Using ARCHIE to perform genome-wide scan of *trans*-eQTLs in a non-trait specific manner can be challenging, as non-trait specific p-value is not computable in current implementation of the method and it will be extremely computational challenging (due to the computational intensive resampling procedure and difficulty of manipulating whole-genome LD matrices⁴). Second, trans-PCO and ARCHIE are designed to capture different *trans* regulatory effects. ARCHIE is powerful when multiple trait-associated variants have weak effects on a single gene (for example, multiple GWAS loci converge on the core genes through *trans* regulation; Figure 2A below and Figure 2 of the ARCHIE paper by Dutta et al.⁴). In contrast, trans-PCO is designed to capture weak *trans* signals of a variant on multiple co-regulated genes, for example, a transcription factor has *trans* effects on multiple target genes (Figure 2B below). Third, *trans* signals captured by ARCHIE are hard to be used to study *trans* regulatory mechanisms. ARCHIE identifies components, consisting of multiple trait-associated SNPs and multiple genes, where sets of gene expressions are *trans* regulated by sets of trait-associated variants. Without knowing the exact *trans*-eQTL SNP driving the *trans* regulation, it is hard to explore the mechanism of the *trans*-eQTL, for example, whether the *trans*-eQTL is also a *cis*-eQTL, or which gene is the *trans* regulator. Fourth, ARCHIE takes all genes as input and infers gene sets as gene components, whereas trans-PCO is flexible to be applied to any user-defined gene set of interest.

A. ARCHIE identifies weak *trans*-eQTL effects from multiple trait-associated SNPs to genes

B. trans-PCO identifies weak *trans*-eQTL effects from a single variant to multiple co-regulated gene

Figure 2. ARCHIE and trans-PCO are designed to detect different *trans*-eQTL signals. (A) ARCHIE detects weak *trans*-eQTL effects from multiple trait-associated SNPs to genes, for example, multiple GWAS SNP to a “core” gene or a “core” process. (B) trans-PCO identifies weak *trans*-eQTL effects from a single SNP to multiple co-regulated genes, for example, from a master regulator to multiple regulated genes.

The differences in goals and usages of ARCHIE and trans-PCO make them not directly comparable. Yet, to give readers and users insights on when and how each method should be used, we thoroughly compared ARCHIE and trans-PCO in both simulations and real data analyses with the following three strategies -

(1) In simulations, we evaluated how well ARCHIE can detect regular *trans*-eQTL signals between a single SNP and a gene set. We used a real co-expression gene module consisting of 101 genes from DGN dataset to simulate data. We simulated z-scores from a normal distribution using the correlation matrix of the gene module, sample size 500, causal gene proportion 30% with 30 genes being true target genes, and genetic variance 0.001 as parameters. More detailed simulation procedure was described in Methods “Simulation”. Trans-PCO has a power of 36% under this setting.

To run ARCHIE, three main inputs are needed⁴, including (1) Σ_{GG} column-correlations of genetic variants, (2) Σ_{EE} column-correlations of gene expressions, and (3) Σ_{GE} cross-covariance matrix between variants and gene expressions. In our simulation settings, we tested one variant at a time. Therefore, $\Sigma_{GG} = 1$. We set $\Sigma_{EE} = \Sigma_{101}$ and we approximate Σ_{GE} with Z_{101}/\sqrt{N} . We used various genetic variances ranging from 0.002 to 0.006. For each scenario, 1000 simulations were performed.

ARCHIE calculated canonical correlations (cc-values, Figure S19A, see below) that measure the *trans* association between each variant and gene sets across 1000 simulations and scenarios. ARCHIE also selected genes (Figure S19B) from the 101 input genes to be target genes in ARCHIE components. While only 30% of genes have true *trans* effects, ARCHIE selected nearly all 101 genes in the ARCHIE component (Figure S19B). To calculate the p-value of a cc-value,

we simulated an empirical null distribution of cc-values by simulating one million null z-scores. Then, an empirical p-value is calculated to be the expected number of null cc-values larger than the observed cc-value (Figure S19C). We found that ARCHIE p-values are deflated across genetic variances, and it was not able to identify significant tests in any of 1000 simulations in our simulation settings, indicating ARCHIE's lower power to detect *trans*-eQTL signals from a single SNP to multiple co-regulated genes (Figure S19D).

Figure S19. Comparison of trans-PCO and ARCHIE in Dutta et al.⁸. (A)-(D) Simulation comparison. (A) Distribution of ARCHIE cc-values across various genetic variances. (B) Number of selected genes by ARCHIE across various genetic variances. Red line indicates the number of true target genes, i.e. 30. (C) QQ plot of empirical p-values across various genetic variances. (D) Distribution of adjusted p-values across various genetic variances. Red line shows FDR level 0.05.

In the Dutta et al. manuscript, they performed simulations to evaluate the performance of ARCHIE, following models similar to Figure 2A (above). However, the simulation was complicated and no source code was released. Therefore, we did not apply trans-PCO to their simulations, instead we came up with two strategies to evaluate how well trans-PCO could replicate *trans*-eQTL signals identified by ARCHIE in real data analyses of eQTLGen.

(2) In eQTLGen, we evaluated how well trans-PCO can replicate signals detected by ARCHIE. ARCHIE identified gene sets that are significantly associated with disease-associated variants of 29 traits using eQTLGen summary statistics, though only results for three traits were publicly available. There were five significant ARCHIE components identified for the three traits. Specifically, 2 (resp. 1 and 2) components were identified to have significant *trans* associations

with prostate cancer (resp. schizophrenia and ulcerative colitis)–associated variants. Each component contains a set of SNPs and a set of genes, and the set of SNPs are correlated to the set of genes through *trans* regulation. We applied trans-PCO to eQTLGen summary statistics and used the five ARCHIE-selected gene sets as gene modules. We performed trans-PCO on the five gene sets and calculated their association p-values with each single eQTLGen variant. P-values were adjusted for multiple testing to control the false positive rate by Bonferroni correction (FDR < 0.05). The goal is to evaluate whether trans-PCO can replicate the *trans*-eQTL signals identified by ARCHIE.

Among the five components, four had significant *trans*-associations by trans-PCO with at least one of ARCHIE selected variants in the same component (Figure S19G, below). Therefore, trans-PCO replicated 80% of the *trans*-signals identified by ARCHIE at 5% FDR. In total, ARCHIE identified 134 variants in the five components. Trans-PCO replicated a large proportion (min:36.5%~max:85.1%) of the variants for component 1's (C1's) for the three diseases (Figure S19G). Only one out of the thirteen variants (7.7%) were replicated by trans-PCO for component 2 (C2) of prostate cancer. These results indicate that trans-PCO is well powered to replicate *trans* signals in component 1's of ARCHIE components, but not component 2's. Nonetheless, trans-PCO identified 1655 additional significant *trans*-eQTL SNPs at 5% FDR, ranging from 65 to 640 for each gene set (Figure S19F).

Figure S19. Comparison of trans-PCO and ARCHIE in Dutta et al.⁸. (E)–(G) Trans-PCO on ARCHIE selected gene sets. (E) X-axis shows significant ARCHIE components for three traits, prostate cancer (Prosc), Schizophrenia, and Ulcerative Colitis (UC). C1 and C2 mean the first and second component. Each component is a pair of selected genes set and variants set. Y-axis shows the size of ARCHIE gene sets (light) and the number of independent null variants (dark) used to estimate correlation matrix of the gene sets. (F) P-values (with Bonferroni correction) of

each variant and ARCHIE gene set by trans-PCO. Each panel is an ARCHIE component. Red cross indicates variants selected by ARCHIE. Grey line shows FDR level 0.05. (G) Number of ARCHIE variants across components that are also significant by trans-PCO (dark).

(3) We directly compared *trans* signals reported by trans-PCO and ARCHIE in the eQTLGen datasets. In summary, ARCHIE identified gene sets that are significantly associated with disease-associated variants of 29 traits, though only results for three traits were publicly available. There were five significant ARCHIE components for the three traits. We also applied trans-PCO to eQTLGen summary statistics as described in the section

“Summary-statistics-based trans-PCO identified 10,167 *trans*-eSNP-module pairs in eQTLGen” in the original manuscript. We analyzed 129 co-expression gene modules and identified 8116 significant *trans*-eSNP-gene co-expression module pairs, corresponding to 2161 eQTLGen test SNPs and 122 gene modules.

To check if ARCHIE signal components found in eQTLGen were replicated by trans-PCO, we checked (1) if ARCHIE selected genes are included in the significant *trans* target gene modules identified by trans-PCO (Figure S19H, below), (2) if ARCHIE selected variants are replicated as *trans*-eQTL SNPs by trans-PCO (Figure S19I). Among selected genes by ARCHIE, all of those included in our eQTLGen analysis were included in a significant *trans*-eQTL module by trans-PCO (Figure S19H). Among selected variants by ARCHIE, 31% (40 out of 129 included variants) were also replicated as significant *trans* signals by trans-PCO (Figure S19I). Additionally, trans-PCO identified 15x more significant *trans*-eQTL SNPs than ARCHIE. These observations support that while both ARCHIE and trans-PCO can identify weak *trans*-effects, they are designed to detect different *trans* signals; ARCHIE is designed to identify the *trans* signals in which a target gene has weak *trans*-effects from multiple SNPs (for example from multiple GWAS SNPs to a single core gene) as modeled in Dutta et al., but trans-PCO is designed to detect *trans* effects from one SNP to multiple genes (for example the *trans* effects from a master regulator to multiple downstream genes).

Figure S19. Comparison of trans-PCO and ARCHIE in Dutta et al.⁸. (H)-(I) Comparison of eQTLGen signals by trans-PCO and ARCHIE. (H) Number of selected genes across ARCHIE components. Lightest blue (Selected) represents the number of selected genes by ARCHIE of each component. Darker blue (Included) represents the selected genes included in trans-PCO eQTLGen analysis. Darkest (Signal) represents genes included in a significant *trans* target gene module by trans-PCO. All ARCHIE selected genes that are analyzed by trans-PCO are in significant *trans* gene modules by trans-PCO. (I) Number of selected variants across ARCHIE components. Labels are similar as in (H).

In summary, we compared trans-PCO to ARCHIE in both simulations and in real data analyses. Our comparison results demonstrate that trans-PCO and ARCHIE are powered at detecting different types of weak *trans*-eQTL signals. For example, ARCHIE has no power to detect weak *trans* effects from one SNP to multiple genes in the simulation analyses; while trans-PCO detects a lot more *trans*-eQTL SNPs for the same gene sets identified by ARCHIE in eQTLGen, it only replicate part of the *trans*-eQTL SNPs selected by ARCHIE. There are also other differences between ARCHIE and trans-PCO. For example, ARCHIE is disease specific and the main goal is to identify *trans* genes associated disease-associated variants, whereas trans-PCO is not disease specific and can be used to perform genome-wide scans of *trans*-eQTLs and produce comprehensive catalogs of *trans*-eQTLs in various tissues and cell types. Furthermore, trans-PCO can be applied to identify *trans*-eQTL SNPs of any user-defined gene sets; in contrast, ARCHIE takes all genes as input and infers a subset of genes *trans* regulated by the variants.

To reflect the revisions, we added more content about ARCHIE in the Introduction (Page 3, Lines 108-111), Results (Page 7, Line 251-259), and Discussion in the main text (Page 18, Lines 593-636). We also included all the detailed comparisons performed in simulations and read data in Supplementary Note (“Compare trans-PCO and ARCHIE”, Page 5-10, Lines 149-329). We also added Figure S19 as a Supplementary figure.

3) One major limitation of the approach is, as acknowledged in the discussion, is that it does not detect individual genes in a network which is driving the *trans*-association signal. I feel the user, however, can use some exploratory tool to prioritize genes in a network that are key drivers for the underlying signal.

Response We thank the reviewer for this great comment and suggestion. We agree with the reviewer that one major limitation of our approach is that it doesn't identify individual genes that are the actual *trans* target, and there can be exploratory ways to prioritize genes. For example, the users can examine the univariate association p-values between the *trans*-eQTL SNP and each gene in the network, and the genes with the most significant p values are likely to be the univariate p values of the genes to estimate the proportion of genes that have π_1 true *trans* effects.

key drivers of the underlying signal. Additionally, users can also use the statistics on the network. These exploratory tools can be extremely useful for the users to interpret the *trans* signals.

Therefore, to reflect the revisions, we added more details to Discussion (Page 19, Lines 647-652), which now reads:

“A limitation of multivariate association tests, including trans-PCO, is that they do not explicitly identify which genes in the gene sets are significantly associated with the test SNP. While functional annotations of gene sets facilitate our understanding of the *trans*-eQTL signals, it is possible that the genes driving *trans*-associations are different from the genes driving functional enrichment of the gene sets. Therefore, the biological interpretation of *trans*-eQTL signals should be supported with other evidence before it is considered definitive. *However, there are exploratory analyses that can help prioritize genes in the network that are key drivers of the underlying signal. For example, by examining the univariate association p-values between the trans-eQTL SNP and each gene in the network, the user can prioritize genes with the most significant p-values as univariate p-values to estimate the proportion of genes that have true π_1 trans effects in the likely trans targets. Furthermore, the users can also use the statistics on the network.*”

4) For trans-loci that colocalize with complex traits, it would be of further interest to explore evidence of causality of the identified gene expression network on the complex trait using the MR framework. Here cis-eQTLs for the genes in the various modules could be used as the underlying instruments for testing evidence of their causal effects on the complex trait. As there are many genes in a module, there needs to be a way of first generating a set of top gene list which is driving the trans-signals (see my comment 3). Then one could test whether cis-eQTL for these top genes show any evidence of association with the underlying complex trait. Being able to show identified gene-networks have causal effects on complex traits will have stronger downstream impact of this work such as for drug development or repurposing, which is one of the ultimate translational applications of all these types of work.

Response We are grateful for the comment and suggestion about using the MR framework to further strengthen the biological support of the network. Indeed, we considered using the MR framework to support the causal role of the colocalized network in complex traits while conducting our study. We used the trans-eQTL SNPs as instrument variables. However, we decided not to continue the work for two reasons: first, trans-PCO associations report p-values, but not the sign of effect. The MR inference without aligning the sign of effects is not robust; second, the horizontal pleiotropy assumption of MR may not hold, where the *trans*-eSNPs might have weak associations with genes outside the gene module. We are therefore thankful to the reviewer for suggesting

using the *cis*-eQTL of the genes in the network as instrument variables and establishing causal effect for each gene and the complex trait.

Transcriptome-wide association study (TWAS) is essentially a MR method with multiple *cis*-SNPs as instrument variables, and thus we used TWAS to find how each gene in the network is associated with the trait. We used this approach to examine the significant colocalization between *trans*-eQTLs of Module 4 and platelet traits. We found existing TWAS studies on platelet traits⁶ (Rowland et al. 2022, Human Molecular Genetics), and there were 1339 unique genes significantly associated with platelet traits. The Module 4 genes (N=507 genes) are significantly enriched in the TWAS significant genes (overlap=88 genes, $p=1.11 \times 10^{-9}$, Fisher's exact test), which further support the role of Module 4 in platelet traits.

Therefore, to reflect the revisions, we added this analysis to Results (Page 14, Lines 454-458), which now reads:

“Additionally, we evaluated whether M4 genes are significantly enriched in genes associated with platelet traits, identified by transcriptome-wide association studies (TWAS). There are 1339 unique genes significantly associated with platelet traits in the UK Biobank. M4 genes are significantly enriched in TWAS genes associated with platelet traits (88 overlap genes, $p\text{-value}=6.7 \times 10^{-10}$, Fisher's exact test), which further supports the role of M4 in platelet traits.”

Similarly, in another colocalization signal between the *trans*-eQTLs of HEME metabolism hallmark gene set and red blood cell traits, we also found significant enrichment of the gene sets in significant TWAS genes associated with red blood cell traits, which further support the role of the gene set in red blood cell traits. We also added this analysis to Results (Page 15, Lines 469-472), which now reads:

“We found that the genes in the gene sets are significantly enriched in TWAS significant genes associated with hemoglobin levels in the UK Biobank (35 overlap genes, $p\text{-value}=8.1 \times 10^{-4}$, Fisher's exact test), which further supports the role of the hallmark gene set in red blood cell traits.”

5) I would like to see a stronger discussion at the end about the translational applications of the proposed work on complex trait genetics (e.g drug development, risk prediction etc).

Response We sincerely thank the reviewer for the comment. We completely agree that understanding *trans* gene regulation is not only important for gaining a fundamental understanding of the complex trait genetics, but also has important translational applications.

To show the translational application of the method, we examined whether drug targets of a disease are more likely to be associated with disease-relevant pathways or gene sets through *trans* regulation. We focused on allergy, because it is immune-related (our gene expression dataset is from blood tissue) and has a relatively large number of drug target genes that are launched (55 launched drug target genes in The Broad Institute Drug Repurposing Hub <https://repo-hub.broadinstitute.org/repurposing>). 5 of 55 drug targets are near allergy associated SNPs in the eQTLGen dataset, therefore tested for *trans* association. We then examined whether these 5 drug targets have significant *trans* associations with immune-related gene co-expression modules or hallmark gene sets in the eQTLGen dataset. Among a total of 142 gene sets (129 co-expression gene modules and 11 hallmark gene sets) used in eQTLGen, 19 were defined as immune-related. Interestingly, we found that all 5 drug target genes are near loci associated with an immune-related gene set through *trans* regulation (Table S19). While the enrichment is not statistically significant ($P=0.12$, Fisher's exact test, Table S20), it is likely due to the small number of drug target genes included in the study. We also note that the *trans-regulated* gene sets of the drug targets are highly relevant to allergy. For example, B cell plays a central role in allergy, and B cell receptor signaling is also enriched in our gene sets. These results support the potential translational application of our method.

Table S19. Allergy drug target genes and their *trans* associated immune-related gene sets

Drug target gene	Gene set type	Gene set index	Selected gene set annotation (see full annotations in Table S7 and Table S15)
SLC37A4	Co-expression module	M54	B cell receptor signaling pathway
UGT3A1	Co-expression module	M54;M62;M76;M87	B cell receptor signaling pathway;TNFR1-induced NFkappaB signaling pathway;NF-kappa B signaling pathway;regulation of T cell activation
IL3	Co-expression module	M54;M108	B cell receptor signaling pathway;Antigen processing and presentation
IL3	Hallmark gene set	HALLMARK_IL6_JAK_STAT3_SIGNALING; HALLMARK_NOTCH_SIGNALING	Genes up-regulated by IL6 via STAT3, e.g., during acute phase response;Genes up-regulated by activation of Notch signaling
ATP5B	Hallmark gene set	HALLMARK_IL6_JAK_STAT3_SIGNALING	Genes up-regulated by IL6 via STAT3, e.g., during acute phase response
FGF1	Hallmark gene set	HALLMARK_IL6_JAK_STAT3_SIGNALING	Genes up-regulated by IL6 via STAT3, e.g., during acute phase response

Table S20. Enrichment of allergy drug targets in *trans* loci associated with immune-relevant gene sets

Fisher's exact test P: 0.12		
	Allergy drug targets near allergy loci in eQTLGen	Non-allergy drug targets near loci in eQTLGen
Genes near trans -eQTL of immune-related gene sets	5	6054
Genes near trans -eQTL of non-immune-related gene sets	0	3180

Additionally, network effects of disease variants can be used for repurposing existing drug compounds to new diseases. Drug repurposing can substantially reduce cost and time to develop new treatments. If the gene expression profiles of an existing drug is enriched for genes in the *trans*-network of another disease's associated loci, it can serve as an evidence for repurposing. Lastly, knowing the network effects of a gene can also help evaluate the safety of a potential drug target. Therapeutic perturbation of a drug target can affect expressions of many downstream genes. While some of them are in the desired disease pathways, others are in pathways associated with other phenotypes, inducing unwanted side-effects.

We are excited about these results and therefore included the results and more discussion in the Discussion section of the main text (Page 20, Lines 686-716), which now reads,

"Identifying the network effects of genetic variants not only shed light on molecular mechanisms of complex associated loci, it can also have important translational applications, for example, in drug discovery and development. First, genes that are associated with disease relevant pathways can serve as evidence for therapeutic targets of the disease. In a preliminary analysis, we examined whether allergy drug targets are more likely to be associated with immune-related gene sets. Among a total of 142 gene sets (129 co-expression gene modules and 11 hallmark gene sets) used for *trans*-eQTL identification in eQTLGen, 19 were defined as immune-related. We used 55 launched allergy drug target genes from The Broad Institute Drug Repurposing Hub (<https://repo-hub.broadinstitute.org/repurposing>), 5 of which are near allergy associated loci in eQTLGen. Interestingly, we found all 5 targets to be associated with immune-related gene sets (Table S19). Detailed analyses can be found in Supplementary Note. While the enrichment is not statistically significant (P=0.12, Fisher's exact test; Table S20), it is likely due to the small number of drug targets included in our analyses.

Additionally, we observed that the *trans* gene modules of drug targets converge to gene sets whose functions are highly relevant to allergy. For example, three drug targets (*IL3*, *UGT3A1* and *SLC37A4*) are associated with gene sets enriched for the B cell signaling pathway. More comprehensive analyses are beyond the scope of this study, yet our preliminary analyses have demonstrated that one can consider genes that have strong *trans* associations with disease-relevant pathways to identify drug targets for the specific disease, especially those with known disease-relevant pathways. Second, network effects of disease variants can be used for repurposing existing drug compounds to new diseases. Drug repurposing can substantially reduce cost and time to develop new treatments. If the gene expression profiles of an existing drug is enriched for genes in the *trans*-network of another disease's associated loci, it can serve as an evidence for repurposing. Additionally, knowing the network effects of a gene can also help evaluate the safety of a potential drug target. Therapeutic perturbation of a drug target can affect expressions of many downstream genes. While some of them are in the desired disease pathways, others are in pathways associated with other phenotypes, inducing unwanted side-effects. We believe comprehensive catalogs of *trans*-networks effects in human cell types and tissues will serve as important resources for interpretation of *trans* regulatory effects of disease associated loci as well as translation applications. Therefore, we made all the trans-PCO *trans*-eQTL signals, with functional annotation of the gene sets, publicly available, downloadable and browsable in www.networks-liulab.org/transPCO."

We also added a section "Drug targets are associated with immune-related gene sets in *trans*" in Supplementary Note (Page 11, Lines 399-418), which now reads,

"To show the translational application of trans-PCO results, we examined whether drug targets are more likely to be associated with disease-relevant pathways or gene sets in *trans*. We first downloaded drug targets of various diseases from The Broad Institute Drug Repurposing Hub (<https://repo-hub.broadinstitute.org/repurposing>). We focused on the disease allergy, because it is immune-related given our analyzed gene expression datasets are from blood tissue. It has a relatively large number of drug target genes (55 launched targets), 5 of which are near (within 1Mb) allergy associated SNPs in eQTLGen (~10k SNPs used for *trans* analysis). We identified SNPs that are significantly associated with allergy using allergy GWAS summary statistics¹¹ (Table S8, p-value<5e-8). We then examined whether these 5 drug targets are near any SNPs that have significant *trans* associations with immune-related gene co-expression modules or hallmark gene sets in the eQTLGen dataset. Among a total of 142 gene sets (129 co-expression gene modules and 11 hallmark gene sets) used in eQTLGen analysis, 19 were defined as immune-related. Interestingly, we found that all 5 drug target genes near allergy loci are associated with an immune-related gene set through *trans* regulation. Details of the targets and their associated immune-relevant gene sets can be found in Table S19. While the enrichment of allergy drug targets in *trans*-eQTLs of immune-related gene sets is not statistically

significant (Table S20), it is likely due to the small number of drug targets in the analyses. Additionally, it is encouraging to see that the gene sets associated with the drug targets are highly relevant to allergy, for example, B cell receptor signaling pathway is associated with three of the drug targets.

We also added Table S19 and Table S20 as Supplementary Tables.

Reviewer #3

This paper proposed trans-PCO, a principal component-based multivariate association test for trans-eQTL. From the simulation studies and real data analysis, the authors showed that the multivariate approach can be more powerful than the univariate approach. Multivariate tests in genetics and genomics have been studied extensively, and it seemed like this a nice application of the multivariate test. I have some comments.

1. The multi-variate test is a nice approach to improve power; however, it is not always more powerful than a single variate test. If the assumption is not true (ex., High level of sparsity, low-level of correlation among variates), it can have a lower power than a single variate test. In the current simulation, since the effect sizes are small, all methods have very low power when the sparsity level is high (low causal proportion). The authors need to consider a situation of high-level sparsity and high effect size (like change sparsity level in Supplementary Figure S1)

Or also can consider a larger sample size (if you think increasing effect size in high sparsity setting is artificial), as some studies (like eQTLGen) have quite a large sample size.

Response We thank the reviewer for this helpful comment. We strongly agree that there are cases when the multivariate test is not more powerful than the univariate test. In our current simulations of high sparsity level, the power of all methods are very low given small effects and therefore can not be easily compared. Therefore, as the reviewer suggested, we performed additional simulations where the sparsity level is high and effect sizes are significantly increased to compare multivariate and univariate methods.

The simulation was performed following the procedure described as in Methods (“Simulation”). Specifically, we used a realistic gene module and the corresponding correlation matrix from the real RNA-seq data consisting of 101 genes ($K = 101$). We simulated 10k z-scores of SNPs from the alternative distribution,

$$Z_{Alt} \sim N(\sqrt{n}[\beta_{101\gamma}, 0]^T, \Sigma_{101}),$$

where n is the sample size, β is a 101γ -long vector representing the causal effect of a SNP on 101 genes, and γ is the proportion of true target genes in the module. Each component of β follows $N(0, \sigma_b^2)$ where σ_b^2 is the genetic variance. We set the sample size n to be 500. We 10%, and the

genetic variances to be as large as 0.01, 0.05, 0.1, and 0.2. We simulated 10k simulated the proportion target genes with non-zero effects to be as low as 1%, 5%, and SNPs and performed 1000 simulations. To control the false discovery rate, we corrected the p-values for multiple testing based on the simulated empirical null distribution of p-values. An association is significant if its adjusted p-value is lower than 0.1. The power is calculated as the proportion of SNPs that were identified to be significant among 10k SNPs. We performed a Wilcoxon test to compare if the mean powers of multivariate and univariate tests are significantly different under each simulation scenario.

Simulation results and comparison of multivariate and univariate tests are added as Figure S1B. Please see below for the added figure in revisions. We observe that univariate method (“MinP”) only has higher power than multivariate method (“trans-PCO”) when the sparsity level is extremely high (i.e. 1% causal out of 101 genes) at large effect sizes. At higher causal proportions, the power of trans-PCO is comparable or higher than MinP at all simulations. These results indicate that MinP has higher power only when the sparsity and the genetic effects are extremely high. Trans-PCO gains more power than MinP when there are more than 1 causal target genes, as it aggregates multiple weak effects to improve power. Additionally, the high sparsity and high effect *trans* signals are not what trans-PCO is designed to detect. Such signals are biologically difficult to interpret, as a single gene does not represent the function of the entire gene set.

To reflect the revisions, we added simulation results to the Results section in main text (Page 7, Lines 232-237). We also added a Figure S1B as a Supplementary figure. The text now reads:

“Simulation results at various genetic variances can be found in Supplementary Materials, including at extremely low proportions of causal genes and high *trans* effects (Figure S1). We found that the univariate method only outperforms trans-PCO when the proportion of causal genes is extremely low, such as only one causal gene in the entire gene set, and the *trans* effects are large. Trans-PCO gains more power when there are more than 1 causal gene, as it aggregates multiple weak effects to improve power. “

Figure S1B. Various genetic variances at extremely low proportions of causal genes. Simulation scenarios when univariate test can be more powerful than multivariate test trans-PCO. We compared the power of trans-PCO and MinP methods under large effect sizes with high levels of sparsity. Specifically, we simulated the proportion of target genes with non-zero effects to be 1%, 5%, and 10%, and large genetic variances to be 0.01, 0.05, 0.1, and 0.2. We used the same gene module as in Figure 2 and simulated the sample size to be 500. Power was computed from 10k SNPs across 1000 simulations. P-values are from the Wilcoxon test to compare two group means. We observe that univariate method (“MinP”) has significantly higher power than multivariate method (“trans-PCO”) when the sparsity level is high and effect sizes are large. For example, in the case of large genetic variance (varb=0.05) and one gene being causal (casual proportion 1%), MinP has significantly higher power than trans-PCO (p-value=1e-8). As causal genes increase, both MinP and trans-PCO have increased power. To be noted, in the case of small genetic variance (varb=0.01) with weak effects, trans-PCO gains more power than MinP as it aggregates multiple weak effects to improve power.

2. As the authors discussed, after the multivariate test, one of the important issues is to identify the gene-snp pairs that drive the signal. I am wondering whether the single variate test can help to figure this out.

Response We thank the reviewer for the comment and suggestion. Indeed, there can be exploratory ways to prioritize genes, such as using the single variate tests. For example, the users can examine the univariate association p-values between the *trans*-eQTL SNP and each gene in the network, and the genes with the most significant p values are likely to be the key p values of the genes to estimate the proportion of genes that have π_1 true *trans* effects. These drivers of the

underlying signal. Additionally, users can also use the statistics on the univariate exploratory tools can be extremely useful for the users to interpret the *trans* signals.

Therefore, to reflect the revisions, we added more details to Discussion (Page 20, Lines 647-652), which now reads:

“A limitation of multivariate association tests, including trans-PCO, is that they do not explicitly identify which genes in the gene sets are significantly associated with the test SNP. While functional annotations of gene sets facilitate our understanding of the *trans*-eQTL signals, it is possible that the genes driving *trans*-associations are different from the genes driving functional enrichment of the gene sets. Therefore, the biological interpretation of *trans*-eQTL signals should be supported with other evidence before it is considered definitive. *However, there are exploratory analyses that can help prioritize genes in the network that are key drivers of the underlying signal. For example, by examining the univariate association p-values between the trans-eQTL SNP and each gene in the network, the user can prioritize genes with the most significant p values as univariate p values to estimate the proportion of genes that have true trans effects in the likely trans targets. Furthermore, the users can also use the statistics on the network.*”

3. Violation of the assumptions in the multivariate test can cause type I error inflation (for example, multivariate normal distribution (MVN) assumption). If the data does not follow MVN, can the method control type I error rates?

Response We thank the reviewer for bringing this concern out. We agree that type I error of a test can be inflated if assumptions of the test are violated. The assumption for PCO is that the z-scores follow normal distribution and the z-scores of multiple traits follow multivariate normal distribution. The gene expression data are inverse rank normalized across individuals before any association analyses. The association test should be well calibrated for each gene and the z-score of each gene follows a normal distribution. Additionally, several literatures^{3,7,8} have supported that the z-scores of association tests of multiple correlated traits asymptotically follow a multivariate normal distribution, with a covariance matrix that is equal to the correlation matrix of phenotypes conditional on covariates. Since this is a common assumption that z-scores of correlated traits follow a multivariate normal distribution (MVN), we do not think it should be a major concern.

4. One of the advantages of the proposed approach the authors mentioned in the paper is the careful removal of misaligned reads.

“we filtered out the reads that were mapped to multiple genomic regions and reads with >2 mismatches.”

But in summary statistics-based analysis (eQTLGen),

“we removed the genes in a module that are cross-mappable with any cis-genes (genes within 100 kb) of the test SNP.”

It doesn't look like they are consistent, as the second seems more like removing cis-effects. The authors need to provide a more detailed explanation of why the second (summary-based) QC is good enough.

Response We thank the reviewer for pointing out the confusion. The reviewer is correct that we used two different ways to make sure the identified signals are not affected by false positives caused by misaligned reads, in cases when raw data or only summary statistics is available. Misaligned reads may introduce false positive *trans* signals, when a SNP has a true *cis* effect on a nearby gene A, but the reads belonging to gene A are falsely mapped to a *trans* gene B. Two genes are called cross-mappable, when they share sequence similarity and sequencing reads can be multi-mapped to either of them. When raw sequence alignment data are available, we filtered out reads that can be multi-mapped to multiple genomic regions to reduce false positives. However, when raw sequence alignment data are not available, such as eQTLGen, we can reliably reduce false positives by removing *trans*-eQTLs whose *trans*-eGenes are cross mappable to the *cis*-Gene of the *trans*-eSNP. This is a reliable and common practice used in previous studies, including the GTEx studies⁹⁻¹¹. By removing “the genes in a module that are cross-mappable with any cis-genes (genes within 100 kb) of the test SNP”, it is not removing cis-effects, Instead, we are making sure not to test the *trans* association between the SNP and any gene that are cross-mappable to the *cis*-gene of the test SNP, therefore avoiding likely false positive *trans* effects.

We admit that the confusion may arise from the unclear description of the procedure. Therefore, we clarified the confusion by revising the description (Page 26, Lines 875-878), which now reads as follows.

“The eQTLGen summary statistics are marginal z-scores meta-weighted across multiple cohorts. Most z-scores are from studies where the RNA-seq reads with mappability issues were not filtered out before quantifying gene expression profiles. Therefore, directly applying trans-PCO to the summary statistics can lead to false positive signals, *which are driven by the cross-mappability between the genes in the module and the cis-gene of the*

test SNP. In order to reduce false positive trans signals, we removed from the gene module genes that are cross-mappable to the cis-gene (within 100kb) of the test SNP, which is a common practice used in previous studies^{8,27,56}.”

5. TpcO is the minimum p-values of several PC-based test approaches, which is not the p-value. Description of p-value calculation of TpcO will be helpful for readers.

Response We thank the reviewer’s suggestion, and we agree adding a description of the p-value calculation will be helpful. Therefore, we added in the Method section the details about p-value calculation. To calculate the p-value of the final omnibus test statistics,

$$T_{PCO} = \min p_{\{PCMinP, PCFisher, PCLC, WI, Wald, VC\}}$$

we first take inverse-normal transformation of the test statistics transform it to be normally distributed and calculate the p-value as,

$$p_{T_{PCO}} = 1 - P\{\min \Phi^{-1}(p_{\{PCMinP, PCFisher, PCLC, WI, Wald, VC\}}) > \Phi^{-1}(T_{PCO}^{obs})\}$$

where Φ^{-1} denotes the inverse standard normal cumulative distribution function. Then the p-value can be efficiently computed using a multivariate normal distribution of six different PC-based tests estimated as described in Liu et al.³.

To clear confusion, we added more details in the Method section (Page 22, Lines 749-754), which now reads as follows.

“The p-value of PCO test statistics can be computed by performing an inverse-normal transformation of the test statistics,

$$p_{T_{PCO}} = 1 - P\{\min \Phi^{-1}(p_{\{PCMinP, PCFisher, PCLC, WI, Wald, VC\}}) > \Phi^{-1}(T_{PCO}^{obs})\}$$

where Φ^{-1} denotes the inverse standard normal cumulative distribution function. The p-value can be efficiently computed using a multivariate normal distribution as described in Liu et al.²⁵.”

Minor comment

1. Figure 2: I cannot see error bars. Is it because of error bars are too small?

Response Yes, the reviewer is correct that the error bars are too small to be visible. We pointed it out in the caption of Figure 2 in the original manuscript (Page 7, Line 248),

“Error bars representing 95% confidence intervals are plotted, but many are too small to be visible.”

References

1. Kolberg, L., Kerimov, N., Peterson, H., and Alasoo, K. (2020). Co-expression analysis reveals interpretable gene modules controlled by trans-acting genetic variants. *eLife* 9, e58705. 10.7554/eLife.58705.
2. Aschard, H., Vilhjálmsson, B.J., Greliche, N., Morange, P.-E., Trégouët, D.-A., and Kraft, P. (2014). Maximizing the Power of Principal-Component Analysis of Correlated Phenotypes in Genome-wide Association Studies. *Am. J. Hum. Genet.* 94, 662–676. 10.1016/j.ajhg.2014.03.016.
3. Liu, Z., and Lin, X. (2019). A Geometric Perspective on the Power of Principal Component Association Tests in Multiple Phenotype Studies. *J. Am. Stat. Assoc.* 114, 975–990. 10.1080/01621459.2018.1513363.
4. Dutta, D., He, Y., Saha, A., Arvanitis, M., Battle, A., and Chatterjee, N. (2022). Aggregative trans-eQTL analysis detects trait-specific target gene sets in whole blood. *Nat. Commun.* 13, 4323. 10.1038/s41467-022-31845-9.
5. Rotival, M., Zeller, T., Wild, P.S., Maouche, S., Szymczak, S., Schillert, A., Castagné, R., Deiseroth, A., Proust, C., Brocheton, J., et al. (2011). Integrating Genome-Wide Genetic Variations and Monocyte Expression Data Reveals Trans-Regulated Gene Modules in Humans. *PLOS Genet.* 7, e1002367. 10.1371/journal.pgen.1002367.
6. Rowland, B., Venkatesh, S., Tardaguila, M., Wen, J., Rosen, J.D., Tapia, A.L., Sun, Q., Graff, M., Vuckovic, D., Lettre, G., et al. (2022). Transcriptome-wide association study in UK Biobank Europeans identifies associations with blood cell traits. *Hum. Mol. Genet.* 31, 2333–2347. 10.1093/hmg/ddac011.
7. Liu, Z., and Lin, X. (2018). Multiple phenotype association tests using summary statistics in genome-wide association studies. *Biometrics* 74, 165–175. 10.1111/biom.12735.
8. Zhu, X., Feng, T., Tayo, B.O., Liang, J., Young, J.H., Franceschini, N., Smith, J.A., Yanek, L.R., Sun, Y.V., Edwards, T.L., et al. (2015). Meta-analysis of Correlated Traits via Summary Statistics from GWASs with an Application in Hypertension. *The American Journal of Human Genetics* 96, 21–36. 10.1016/j.ajhg.2014.11.011.
9. THE GTEx CONSORTIUM (2020). The GTEx Consortium atlas of genetic regulatory effects across human tissues. *Science* 369, 1318–1330. 10.1126/science.aaz1776.

10. Saha, A., and Battle, A. (2019). False positives in trans-eQTL and co-expression analyses arising from RNA-sequencing alignment errors. *F1000Res* 7, 1860. [10.12688/f1000research.17145.2](https://doi.org/10.12688/f1000research.17145.2).
 11. Battle, A., Mostafavi, S., Zhu, X., Potash, J.B., Weissman, M.M., McCormick, C., Haudenschild, C.D., Beckman, K.B., Shi, J., Mei, R., et al. (2014). Characterizing the genetic basis of transcriptome diversity through RNA-sequencing of 922 individuals. *Genome Res.* 24, 14–24. [10.1101/gr.155192.113](https://doi.org/10.1101/gr.155192.113).
-

Referees' report, second round of review

Reviewer #1:

I previously commented on the lack of comparisons that I perceived, with regards to other methods that aim to do the same thing. The authors have now provided a comprehensive comparison, and describe in their response to the reviewers in length what these differences entail. For me this helps tremendously to understand the novelty of this method. I am now also convinced this paper is novel, and believe it is a valuable addition to the field.

However, I would recommend that the authors would make this extensive answer they provide in the “response to reviewers” available to the scientific audience as well, because the amount of text that has been added to the manuscript is somewhat limited. As such I would request the authors to add this as a “supplementary note”. I hope the authors are willing to do so, since it has helped me tremendously to understand the difference of this method with existing methods. Reviewer 2 also had exactly this question, and as such I suppose this applies to others as well.

Reviewer #3:

The authors addressed my comments. I don't have any additional ones.

Authors' response to the second round of review

We sincerely thank the reviewers for their time and effort reviewing our revised manuscript. We are glad that we addressed all the reviewers' comments by doing comprehensive method comparisons. In response to the reviewer's request, we added the extensive answer in response letter to our manuscript and supplement.

Specifically, We summarize the method comparisons in the manuscript. For example, in Results, it now reads:

“We included comparisons to two additional methods: ARCHIE proposed by Dutta et al.¹⁶ and a method by Rovital et al.¹⁴ (see Supplementary Note, Figure S19 and Figure S20). We showed that ARCHIE is not powerful at detecting *trans*-eQTL effects from a SNP to multiple genes, which are the effects trans-PCO was designed for (Figure S19). We note that the main goal of ARCHIE is to identify trait-specific gene sets associated with GWAS loci, whereas trans-PCO is designed to map *trans*-eQTLs for any user-specified gene sets in specific tissues or cell types (see Supplementary Note, Figure S19 and Discussion). Rovital et al.¹⁴ is based on the primary PCbased approach and we showed that the method has limited power at identifying weak *trans*eQTL effects (Supplementary Note and Figure S20). “

In Discussion, it now reads:

“We thoroughly compared the performance of trans-PCO versus other methods, such as the PC1–based method by Kolberg et al.¹⁵, ARCHIE by Dutta et al.¹⁶ and Rovital et al.¹⁴ (see Supplementary Note, Figure 2, Figure S13-S14, Figure S18-S20). Trans-PCO and the PC1–based method are both designed to identify individual *trans*-eQTLs of any gene sets containing multiple genes, and the comparison between them is straightforward. However, ARCHIE is different and not directly comparable to the other two methods for several reasons (see more discussions in Supplementary Note). First, ARCHIE captures only trait-specific *trans* regulations, by testing significance against a null hypothesis based on a subset of genetic variants that are trait-associated. In contrast, trans-PCO identifies *trans*-eQTLs under the general null hypothesis with no additional assumptions. Second, trans-PCO and ARCHIE are designed to capture different *trans* regulatory effects. ARCHIE is powerful when multiple disease–associated variants have weak effects on a single gene (for example, multiple GWAS variants converge onto the core genes through *trans* regulation) or multiple disease–associated variants have weak effects on multiple genes (Figure 2 in Dutta et al.¹⁶), which are not co-regulated by a shared *trans* genetic locus. In contrast, trans-PCO is designed to capture weak *trans* signals of a variant on multiple co-regulated genes (see Supplementary Notes and Figure S19). Third, ARCHIE identifies components, consisting of multiple trait-associated SNPs and multiple genes, without knowing the exact *trans*-eQTL SNP driving the *trans* regulation. It is hard to further study *trans* regulatory mechanisms of the *trans*-eQTLs. Fourth, ARCHIE takes all genes as input and infers gene sets that are *trans*-regulated by disease-associated variants, whereas trans-PCO is flexible to be applied to any user-defined gene set of interest to identify *trans*-eQTLs. In summary, trans-PCO and ARCHIE have different goals and are designed for detecting different types of *trans* signals. Yet, we thoroughly compared ARCHIE and trans-PCO in both simulations and real data analyses (Supplementary Note). We believe these comparisons will provide insights on when and how these methods should best be used. “

Additionally and importantly, we reorganized the structure of the Supplement Note by adding a new section titled “Method S3. Compare trans-PCO with existing methods”, which includes three sub-sections A) Compare trans-PCO with primary PC method; B) Compare trans-PCO and ARCHIE, and C) Comparison trans-PCO and Rotival et al.. We included all the details in the response letter to each session, including all the data analyses, simulations, the results and the figures in the response letter.

We hope the revised version will help the readers recognize the differences between the methods as well as the technical advance of trans-PCO.

Referees' report, third round of review

Reviewer #1:

I thank the authors for providing a comprehensive supplementary note on the comparisons with existing methods. Thank you!

Authors' response to the third round of review

NA